# VideoBrain: Learning Adaptive Frame Sampling for Long Video Understanding

**Junbo Zou** [1] **Ziheng Huang** [2] **Shengjie Zhang** [3] **Liwen Zhang** [4] **Weining Shen** [3]

## Abstract

Long-form video understanding remains challenging for Vision-Language Models (VLMs) due to the inherent tension between computational constraints and the need to capture information distributed across thousands of frames. Existing approaches either sample frames uniformly (risking information loss) or select keyframes in a single pass (with no recovery from poor choices). We propose VideoBrain, an end-to-end framework that enables VLMs to adaptively acquire visual information through learned sampling policies. Our approach features dual complementary agents: a CLIP-based agent for semantic retrieval across the video and a Uniform agent for dense temporal sampling within intervals. Unlike prior agent-based methods that rely on text-only LLMs orchestrating visual tools, our VLM directly perceives frames and reasons about information sufficiency. To prevent models from invoking agents indiscriminately to maximize rewards, we introduce a behavior-aware reward function coupled with a data classification pipeline that teaches the model when agent invocation is genuinely beneficial. Experiments on four long video benchmarks demonstrate that VideoBrain achieves +3.5% to +9.0% improvement over the baseline while using 30-40% fewer frames, with strong cross-dataset generalization to short video benchmarks. The code is available at https://github.com/junbo-zou/VideoBrain.

---
[1]Georgia Institute of Technology, GA, United States [2]Columbia University, NY, United States [3]University of California, Irvine, CA, United States [4]Shanghai University of Finance and Economics, Shanghai, China. Correspondence to: Liwen Zhang <zhang.liwen@shufe.edu.cn>, Weining Shen <weinings@uci.edu>.

*Proceedings of the 43rd International Conference on Machine Learning*, Seoul, South Korea. PMLR 306, 2026. Copyright 2026 by the author(s).

## 1. Introduction

Video understanding has emerged as a critical capability for Vision Language Models (VLMs), with applications spanning content analysis, autonomous driving, and embodied AI (Lin et al., 2024; Tian et al., 2024; Suglia et al., 2024). However, long-form video understanding presents fundamental challenges: videos can span hours with information distributed sparsely across thousands of frames, yet computational constraints limit the number of frames that can be processed simultaneously. This poses a key challenge: while processing more frames improves understanding, it dramatically increases cost. Critically, the required temporal granularity varies by query—some questions demand dense frame sampling to capture fine-grained dynamics, while others need only a few keyframes. This motivates the need for adaptive strategies that dynamically allocate computational resources based on query complexity.

Existing approaches address this challenge through various frame sampling strategies. Uniform sampling (Maaz et al., 2023; Song et al., 2024; Shen et al., 2024) extracts frames at fixed intervals regardless of content, risking critical information loss when events occur between sampled frames. Adaptive keyframe selection methods (Tang et al., 2025; Yao et al., 2025) improve upon this by selecting frames based on visual diversity or query relevance, but operate in a single pass—once frames are selected, there is no mechanism to recover from poor initial choices. Recent agent-based approaches (Wang et al., 2024c; Zhi et al., 2025) enable iterative information gathering, but employ a decoupled architecture where a text-only LLM orchestrates visual tools without directly perceiving video content. This creates an information bottleneck: the LLM makes sampling decisions based solely on textual descriptions, potentially missing fine-grained visual cues essential for complex reasoning.

To overcome the information bottleneck in long video understanding, we propose VideoBrain, an end-to-end framework that enables VLMs to adaptively acquire visual information through learned sampling policies. Unlike pipeline approaches, our VLM directly perceives video frames and reasons about whether current information suffices for answering. VideoBrain employs two complementary sampling agents: a CLIP-based agent for semantic retrieval across the video (e.g., finding specific visual content regardless of

temporal location) and a Uniform agent for dense temporal sampling within intervals (e.g., understanding sequential actions). The model learns when and how to invoke these agents through a two-stage training process combining supervised fine-tuning and reinforcement learning with behavior-aware rewards.

A key challenge in training agentic VLMs is reward hacking (Zheng et al., 2025; He et al., 2025): models may learn to invoke agents indiscriminately to maximize rewards, even when initial frames already contain sufficient information. We address this through a novel behavior-aware reward function that explicitly discourages unnecessary agent calls for Direct questions (where initial frames suffice) while encouraging exploration for Active questions (where additional visual information is genuinely needed), directly preventing reward hacking. This is enabled by our data classification pipeline that categorizes questions into Direct, Adaptive, and Active based on whether agent invocation provides genuine benefits.

Our main contributions are:

- We propose VideoBrain, an end-to-end trainable framework with dual sampling agents (CLIP-based semantic retrieval and uniform temporal sampling) that enables adaptive frame sampling through iterative reasoning.

- We introduce a novel behavior-aware reward function and data classification pipeline that prevents reward hacking, teaching the model to dynamically calibrate its behavior: invoking agents for complex queries that require additional visual information, while directly answering simpler questions without unnecessary sampling.

- Extensive experiments on four long video benchmarks demonstrate consistent improvements (+3.5% to +9.0%) over the baseline while using 30-40% fewer frames, with strong generalization to short video benchmarks. Comprehensive ablation studies further validate the effectiveness of each component.

## 2. Related Work

### 2.1. Frame Sampling Strategies for Video Understanding

**Uniform Sampling.** Most video understanding models (Maaz et al., 2023; Song et al., 2024; Shen et al., 2024) employ uniform frame sampling, extracting frames at fixed temporal intervals regardless of content or query. While computationally straightforward, this approach fundamentally limits long video understanding: sparse sampling risks missing critical events, while dense sampling incurs prohibitive computational costs. While recent VLMs have

achieved impressive performance on short video benchmarks (Xia et al., 2025b;a), long-form video understanding remains challenging. Recent benchmarks (Fu et al., 2025; Li et al., 2024; Mangalam et al., 2024) consistently demonstrate significant performance degradation as video duration increases, highlighting the inadequacy of content-agnostic sampling strategies.

**Adaptive Keyframe Selection.** To overcome uniform sampling limitations, recent works explore adaptive frame selection. Training-free methods such as AKS (Tang et al., 2025) and AdaRD-key (Zhang et al., 2025) select keyframes based on visual diversity or relevance metrics, while K-frames (Yao et al., 2025) and F2C (Sun et al., 2025) propose plug-and-play modules for question-aware selection. However, these approaches operate in a **single-pass** manner: they select frames once before reasoning, without the ability to iteratively refine selections based on intermediate understanding. When initial selections miss crucial information, there is no mechanism for recovery.

**Agent-Based Video Understanding.** VideoAgent (Wang et al., 2024c) and VideoAgent2 (Zhi et al., 2025) pioneer agentic approaches that iteratively gather information through CLIP-based semantic retrieval. However, they employ a **decoupled architecture** where a text-only Large Language Model (LLM) orchestrates visual tools without directly perceiving video content. This creates an **information bottleneck**: visual information must first be converted to text descriptions, and the LLM makes decisions based solely on these potentially incomplete or inaccurate textual representations. Fine-grained visual details critical for answering complex questions may be lost in this vision-to-text conversion. Furthermore, the multi-stage pipeline prevents end-to-end optimization, and errors from individual components (e.g., inaccurate captions) propagate through the system.

### 2.2. Agentic Reinforcement Learning for Vision-Language Models

**Image Domain.** Recent works have applied reinforcement learning to enable agentic behavior in Vision-Language Models. DeepEyes (Zheng et al., 2025) and GRIT (Fan et al., 2025) train models to dynamically explore images through iterative zooming and region cropping, acquiring fine-grained local information for more accurate responses. These methods demonstrate the potential of learned policies for adaptive visual perception, though they focus exclusively on static images.

**Video Domain.** Extending agentic reasoning to video, FrameMind (Ge et al., 2025), FrameThinker (He et al., 2025), DeepSport (Zou et al., 2025) and Video-

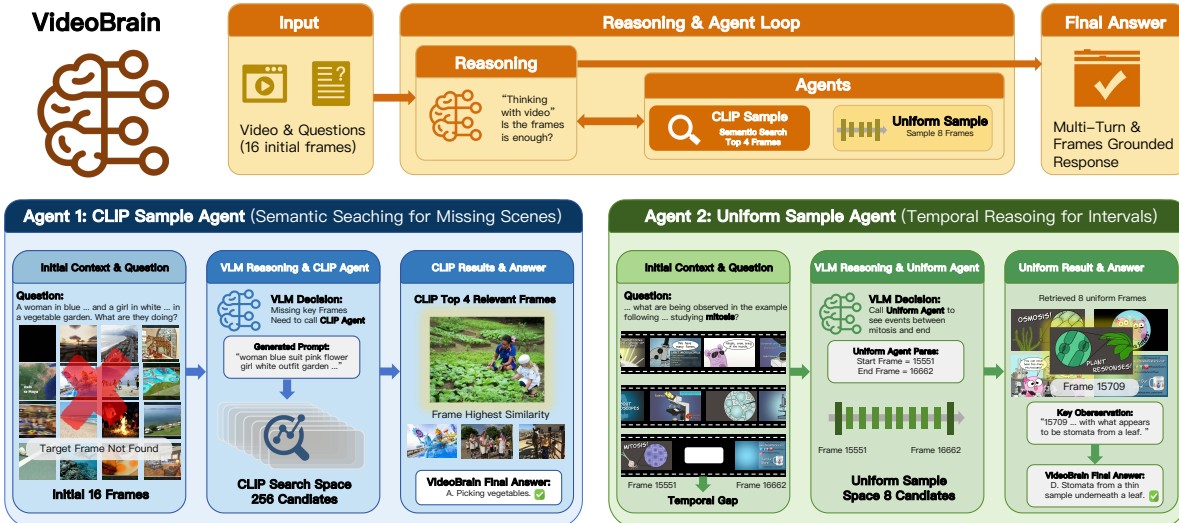

*Figure 1.* VideoBrain framework and two complementary sampling agents. **Top:** The model takes video frames and a question as input, reasons about information sufficiency, and selectively invokes agents to gather additional frames before producing a final answer. **Bottom left:** The CLIP Sample Agent searches semantically across 256 candidate frames to retrieve visually relevant content (e.g., finding a specific scene described in the question). **Bottom right:** The Uniform Sample Agent densely samples within a temporal interval to capture fine-grained sequential information (e.g., observing what happens between two events).

Thinker (Wang et al., 2025) leverage reinforcement learning to train VLMs that interleave reasoning with frame sampling through multi-turn interactions. While promising, these methods share fundamental limitations. First, they are restricted to **uniform temporal sampling** within specified intervals, unable to perform semantic retrieval for visually similar content scattered across the video—when target information appears at unpredictable locations, uniform sampling may repeatedly miss relevant frames. Second, their reward functions typically assign positive rewards for any agent invocation, creating a **reward hacking** problem: models learn to invoke sampling tools indiscriminately, even when initial frames already contain sufficient information. Third, these methods lack explicit mechanisms to determine *when* additional sampling is truly necessary, failing to distinguish between questions answerable from initial context and those genuinely requiring more visual information.

Our VideoBrain fundamentally addresses these limitations through three key innovations. First, we enable **dual sampling strategies**—both semantic retrieval (via CLIP) and uniform temporal sampling—allowing the model to choose the appropriate strategy based on question type. Second, we introduce a **behavior-aware reward function** that explicitly discourages unnecessary agent calls for Direct questions while encouraging exploration for Active ones, directly preventing reward hacking. Third, our **end-to-end architecture** allows the VLM to both perceive video frames and make sampling decisions, eliminating the information bottleneck inherent in pipeline approaches. Unlike prior methods

where a text-only LLM orchestrates visual tools, our VLM directly observes frames and reasons about whether current information suffices.

## 3. Method

We present VideoBrain, an end-to-end trainable framework that enables VLMs to adaptively acquire visual information through learned sampling policies. Our approach consists of two main components: (1) a multi-agent architecture for adaptive frame sampling during inference, and (2) a training pipeline with behavior-aware rewards that teaches the model when and how to invoke sampling agents.

### 3.1. Multi-Agent Architecture

Unlike pipeline approaches where a text-only LLM orchestrates visual tools based on intermediate text descriptions, VideoBrain employs an end-to-end architecture where the VLM directly perceives video frames and makes sampling decisions, avoiding the information bottleneck inherent in vision-to-text conversion.

**Iterative Reasoning and Sampling.** Given a video $V$ and a question $Q$, the goal is to select an optimal frame subset $\mathcal{F}$ that maximizes answer accuracy under a frame budget constraint:

$$\max_{\mathcal{F} \subseteq V} P(y|Q, \mathcal{F}; \theta) \quad \text{s.t.} \quad |\mathcal{F}| \leq N \quad (1)$$

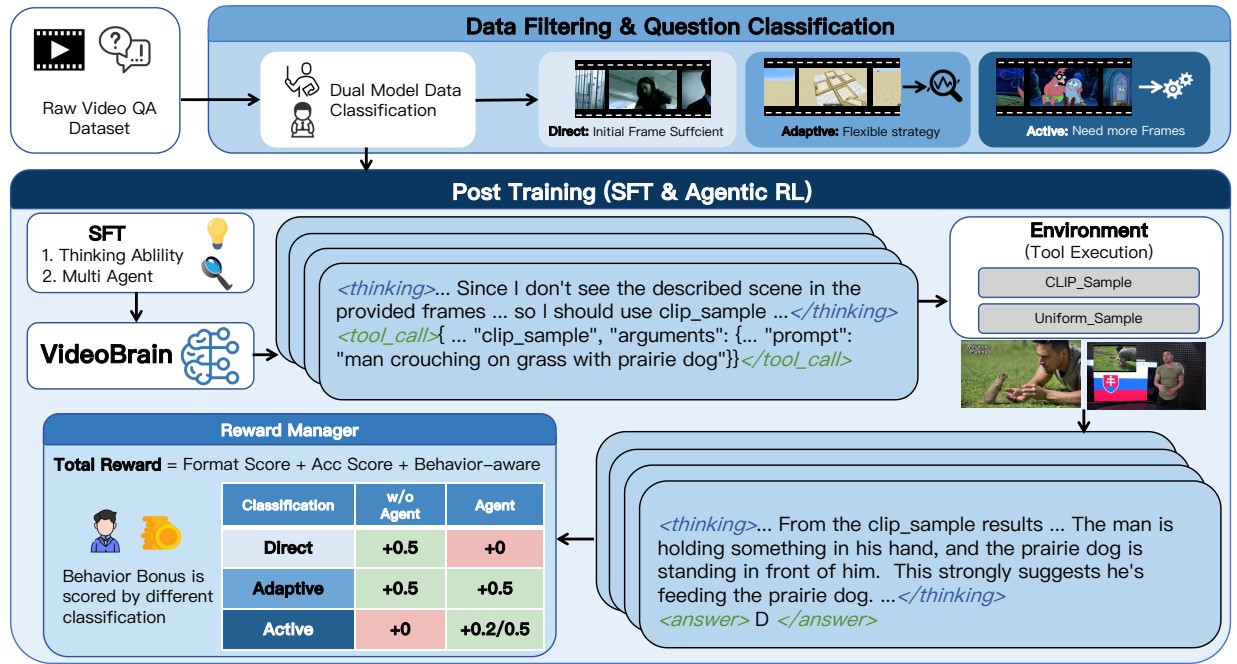

*Figure 2.* Overview of the VideoBrain training framework. **Top:** Dual-model evaluation classifies video QA samples into Direct, Adaptive, and Active categories based on whether agent invocation improves performance. **Bottom:** SFT teaches thinking and agent invocation. During RL, the model iteratively reasons and calls sampling agents to gather frames, with behavior-aware rewards encouraging efficiency on Direct questions and exploration on Active ones.

where $y$ is the correct answer, $\theta$ denotes model parameters, and $N$ is the maximum number of frames. VideoBrain first extracts $N_0$ uniformly sampled initial frames as the starting visual context. The VLM then enters an iterative reasoning loop: it analyzes the current frames, reasons about whether the available information is sufficient to answer the question, and either (1) outputs a final answer if confident, or (2) invokes a sampling agent to acquire additional frames and continues reasoning. This process repeats for at most $K$ interaction rounds, enabling the model to progressively gather visual evidence as needed.

**Output Format.** At each reasoning step, the model generates structured output in one of two forms. For intermediate reasoning, the model outputs `<thinking>...</thinking>` followed by `<tool_call>{...}</tool_call>`, where the tool call specifies which agent to invoke and its parameters. When the model determines it has sufficient information, it outputs `<thinking>...</thinking>` followed by `<answer>...</answer>` containing the final response. This format enables explicit chain-of-thought reasoning while maintaining structured agent interactions.

**CLIP Sample Agent.** The CLIP Sample Agent performs semantic retrieval to locate visually relevant frames across the video. Given parameters (start_frame, end_frame,

prompt), it retrieves frames whose visual content best matches the text prompt, regardless of their temporal position. As shown in Algorithm 1, we first uniformly sample $m$ candidate frames from the specified interval, where $m$ scales with video length (128 for shorter videos, 256 for videos exceeding 20,000 frames). We then compute CLIP similarity scores between the text prompt embedding and each candidate frame's visual embedding, returning the top-$n$ frames ranked by similarity. This agent excels at finding specific visual content scattered across the video, such as "a woman wearing a red coat" or "the moment when the character falls down."

**Uniform Sample Agent.** The Uniform Sample Agent provides temporally dense sampling within a specified interval. Given parameters (start_frame, end_frame), it returns $n$ frames uniformly distributed across the interval. Unlike CLIP-based retrieval, this agent preserves temporal structure and is particularly effective for questions requiring sequential understanding, such as "what happened between event A and event B" or "describe the actions in chronological order." The uniform spacing ensures comprehensive coverage of the temporal segment without semantic bias.

**Complementary Strategies.** The two agents provide complementary capabilities that together address the diverse information needs in video understanding. The CLIP Sample

**Algorithm 1** CLIP-Based Content Retrieval

1: **function** CLIPSAMPLE($f_s, f_e, n, P$)
2: $L \leftarrow f_e - f_s$ {**1. Validation**}
3: **if** $L \leq n$ **then**
4:     **return** ERROR("Invalid segment")
5: **end if**{**2. Candidate Sampling**}
6: $m \leftarrow \begin{cases} L, & \text{if } L < 128 \\ 128, & \text{if } 128 \leq L < 20000 \\ 256, & \text{otherwise} \end{cases}$
7: $C \leftarrow$ UNIFORMSAMPLE($f_s, f_e, m$) {**3. CLIP-Based Scoring**}
8: $\mathbf{t} \leftarrow$ CLIP.TEXT($P$)
9: $S \leftarrow []$
10: **for** $f \in C$ **do**
11:     $\mathbf{v} \leftarrow$ CLIP.IMAGE($f$)
12:     $s \leftarrow$ COSINESIMILARITY($\mathbf{t}, \mathbf{v}$)
13:     $S$.APPEND($(f, s)$)
14: **end for**{**4. Top-n Selection**}
15: $S \leftarrow$ SORT($S$, descending)
16: $R \leftarrow [S[i].f \mid i \in \{1, \ldots, n\}]$
17: **return** $R$
18: **end function**

Agent enables semantic search across the entire video, effectively locating content that may appear at unpredictable temporal locations—essential for questions like "find when the character wears a specific outfit." The Uniform Sample Agent excels at capturing fine-grained temporal dynamics within a localized segment, crucial for understanding action sequences or state transitions. By learning to select the appropriate agent based on question type, VideoBrain can efficiently navigate both the semantic and temporal dimensions of video content.

### 3.2. Data Selection and Classification

**Two-Model Joint Selection.** As illustrated in Figure 2, we employ a two-model approach to classify training samples based on whether agent invocation is beneficial. The **Base Model** receives only the $N_0$ initial uniformly sampled frames and answers questions without agent access. The **Teacher Model**, with larger capacity and stronger reasoning capabilities, has full access to sampling agents and can iteratively gather additional frames. By comparing their performance on each sample, we identify cases where agent usage provides genuine benefit versus cases where initial frames already suffice.

**Question Categories.** Based on the joint evaluation, we classify each question-video pair into three categories:

- **Direct:** Both Base Model and Teacher Model answer correctly. The initial frames contain sufficient informa-

tion.

- **Adaptive:** The Base Model fails while the Teacher Model succeeds without invoking agents, indicating that the smaller model has limited visual perception capability. These samples constitute a small proportion of the training data. We do not enforce a specific strategy for them, as agent invocation may also help improve accuracy in some cases.

- **Active:** The Base Model fails, and either the Teacher Model succeeds by invoking agents, or the Teacher Model also fails. These questions require additional visual information.

Samples where the Base Model succeeds but the Teacher Model fails with agent usage are filtered out as anomalies.

### 3.3. Training

**Supervised Fine-Tuning.** Since the base VLM already possesses basic visual understanding and instruction-following capabilities, we use a small amount of data for cold-start training. We perform SFT using trajectories generated by the Teacher Model on Adaptive and Active samples—both categories where the Base Model fails, providing effective learning signals. These trajectories include cases with and without agent invocations, teaching the model to recognize when additional frames are needed. This stage primarily teaches the model two capabilities: (1) *reasoning ability*—learning to analyze current visual information and determine whether it suffices for answering; and (2) *tool usage ability*—learning when to invoke agents, how to craft effective prompts for the CLIP agent to retrieve semantically relevant frames, and how to specify appropriate intervals for the Uniform agent to capture temporal details.

**Reinforcement Learning.** We employ Group Relative Policy Optimization (GRPO) (Shao et al., 2024) to further refine the model's agent usage policy. For each prompt $q$ from the RL set, we sample a group of $G$ trajectories $\{\tau_i\}_{i=1}^{G} \sim \pi_{\text{old}}(\cdot \mid q)$ and obtain rewards $\{r_i\}_{i=1}^{G}$. The training objective is:

$$\mathcal{J}_{\text{GRPO}}(\theta) = \mathbb{E}\left[\frac{1}{G}\sum_{i=1}^{G}\min\left(\rho_i A_i, \text{ clip}(\rho_i, 1-\varepsilon, 1+\varepsilon)\right.\right.$$
$$\left.\left. A_i\right) - \beta\,\text{KL}\big(\pi_\theta(\cdot \mid q) \,\|\, \pi_{\text{ref}}(\cdot \mid q)\big)\right], \quad (2)$$

where $\rho_i = \pi_\theta(\tau_i|q)/\pi_{\text{old}}(\tau_i|q)$ is the importance ratio, $\pi_{\text{ref}}$ is the reference model (the SFT checkpoint), and $A_i$ is the advantage computed from group statistics. The model is trained on all three question categories (Direct, Adaptive, Active), learning to discriminate when agent invocation is beneficial versus wasteful.

*Table 1.* Behavior-aware reward $R_{\text{behavior}}$ based on question category and agent usage. ✓ denotes correct answer, × denotes incorrect. The omitted column "w/o Agent ×" is zero for all categories.

| Category | w/o Agent ✓ | Agent ✓ | Agent × |
|---|---|---|---|
| Direct | +0.5 | 0 | 0 |
| Adaptive | +0.5 | +0.5 | 0 |
| Active | 0 | +0.5 | +0.2 |

## 3.4. Reward Design

The total reward is computed as:

$$R = \mathbb{I}_{\text{format}} \cdot (R_{\text{format}} + R_{\text{accuracy}} + R_{\text{behavior}}), \quad (3)$$

where $\mathbb{I}_{\text{format}}$ is an indicator function that equals 1 if the output passes format validation, and 0 otherwise.

**Format Reward.** The format reward $R_{\text{format}}$ validates the structural correctness of model outputs. A reward of 0.05 is granted when the output passes all format checks: (1) proper pairing of `<thinking>`, `<tool_call>`, and `<answer>` tags; (2) non-empty content within each tag; and (3) valid JSON format for tool calls without duplicate invocations. If format validation fails, the entire reward is zero ($\mathbb{I}_{\text{format}} = 0$), and $R_{\text{accuracy}}$ and $R_{\text{behavior}}$ are not computed. This design penalizes unstable outputs such as excessive tag repetition or malformed tool calls.

**Accuracy Reward.** The accuracy reward $R_{\text{accuracy}}$ evaluates answer correctness using LLM-as-a-judge. For multiple-choice questions, we use Qwen-Flash (Yang et al., 2025) for strict binary scoring (0 or 1). For open-ended questions, we use DeepSeek-V3.2(DeepSeek-AI, 2025) to assess semantic similarity, yielding scores in $[0, 1]$ based on alignment with ground truth.

**Behavior-Aware Reward.** The behavior-aware reward $R_{\text{behavior}}$ explicitly encourages appropriate agent usage based on question category, as shown in Table 1:

For **Direct** questions where initial frames suffice, the model receives a reward only when answering without invoking agents, encouraging efficiency. For **Adaptive** questions reflecting model capability gaps, both strategies receive equal reward, allowing flexibility. Since a substantial proportion of trajectories in Direct and Adaptive categories can achieve high or full scores, no failure incentive is needed. However, for **Active** questions, a significant portion includes cases where even the Teacher Model fails, making these inherently difficult with sparse rewards. To address this, we provide a partial reward (0.2) for appropriate agent invocation even when the answer is incorrect, encouraging the model to develop exploration behavior rather than avoiding agent calls due to reward sparsity.

This design directly prevents reward hacking: the model cannot gain extra reward by indiscriminately calling agents on every question. Instead, it must learn to adopt the appropriate strategy—being efficient on Direct questions while being thorough on Active ones.

## 4. Experiment

### 4.1. Implementation Details

**Model Configuration.** We use Qwen3-VL-8B-Instruct (Bai et al., 2025a) as the Base Model for training VideoBrain. For data classification, we employ Qwen3-VL-235B-Thinking (Bai et al., 2025a) as the Teacher Model due to its larger capacity and stronger reasoning capabilities. The initial number of frames $N_0$ is set to 16, and the maximum interaction rounds $K$ is set to 5. Each agent call returns 4 frames for CLIP Sample Agent and 8 frames for Uniform Sample Agent. For the CLIP Sample Agent, we use SigLIP2-B/16 (Tschannen et al., 2025) for efficiency, as it provides a good balance between retrieval quality and computational cost.

**Training Data.** We curate training data from multiple video QA datasets: Video-Holmes (Cheng et al., 2025), CG-Bench (Chen et al., 2024a), NExT-QA (Xiao et al., 2021), MLVU (Zhou et al., 2025), and LongVideo-Reason (Chen et al., 2025). After dual-model classification, we obtain approximately 8K samples in total, with 1.6K samples for SFT and 6.4K samples for RL. The SFT set contains 42.3% Adaptive and 57.7% Active samples (no Direct, as these are already answerable by the base model), while the RL set contains 44.9% Direct, 8.1% Adaptive, and 47.0% Active samples. Both stages are trained for 1 epoch.

**Computing Resource.** All experiments are conducted on 8×H20 GPUs. The total training cost is approximately 472 GPU hours.

### 4.2. Main Result

**Benchmarks.** We evaluate VideoBrain on four challenging long video understanding benchmarks. **LongVideoBench** (Wu et al., 2024) contains videos ranging from 1 to 60 minutes with diverse question types. **LVBench** (Wang et al., 2024b) focuses on extremely long videos (30-140 minutes) requiring comprehensive understanding. **Video-MME** (Fu et al., 2025) is a multi-modal evaluation benchmark; we report results on the long video subset (30-137 minutes). **MLVU Test** (Zhou et al., 2025) test set covers videos from 3 to 60 minutes with multi-level understanding tasks.

**Baselines.** We compare with both proprietary models GPT-4o (OpenAI, 2024) and Gemini-1.5-Pro (Deep-

*Table 2.* Comparison with state-of-the-art models on long video understanding benchmarks.

| Benchmark | | LongVideoBench
1-60 min | | LVBench
30-140 min | | Video-MME Long
30-137 min | | MLVU Test
3-60 min | |
|---|---|---|---|---|---|---|---|---|---|
| **Model** | **Params** | **Acc** | **Frame** | **Acc** | **Frame** | **Acc** | **Frame** | **M-AVG** | **Frame** |
| *Proprietary Models* | | | | | | | | | |
| GPT-4o | - | 58.5 | 32 | 48.9 | 60 | 65.3 | 384 | 54.9 | 0.5fps |
| Gemini-1.5-Pro | - | 55.2 | 32 | 33.1 | 3600 | 67.4 | 0.5fps | - | - |
| *Open Source Models* | | | | | | | | | |
| ShareGPT4Video | 7B | 39.7 | 16 | - | - | 37.9 | 16 | 33.8 | 16 |
| LongVA | 7B | - | - | - | - | 46.2 | 128 | 41.1 | 128 |
| VideoChat-R1 | 7B | 49.1 | 32 | 34.3 | 32 | 46.2 | 32 | - | - |
| Video-R1 | 7B | 52.7 | 32 | 35.3 | 32 | 48.2 | 32 | 45.4 | 32 |
| Qwen2.5-VL-Instruct | 7B | 43.2 | 32 | 31.6 | 32 | 41.9 | 32 | 41.6 | 32 |
| FrameThinker | 7B | 52.9 | 21.1 | 36.6 | 23.9 | 47.6 | 23.9 | - | - |
| Qwen3-VL-Instruct | 8B | 45.2 | 32 | 32.3 | 32 | 45.6 | 32 | 43.4 | 32 |
| VideoBrain (Ours) | 8B | 53.3 | 22.3 | 41.3 | 21.6 | 49.8 | 19.1 | 46.9 | 20.0 |
| $\Delta$ vs Baseline | | +8.1 | -30% | +9.0 | -33% | +4.2 | -40% | +3.5 | -38% |

Mind, 2024), as well as open-source models including ShareGPT4Video (Chen et al., 2024b), LongVA (Zhang et al., 2024), VideoChat-R1 (Li et al., 2025), Video-R1 (Feng et al., 2025), and Qwen2.5-VL-Instruct (Bai et al., 2025b). We also compare with FrameThinker (He et al., 2025), a recent agentic RL approach that employs uniform temporal sampling for iterative frame retrieval. Additionally, we include Qwen3-VL-8B-Instruct (Bai et al., 2025a) as our direct baseline since VideoBrain is built upon it.

**Results.** As shown in Table 2, VideoBrain achieves consistent improvements over the Qwen3-VL-8B-Instruct baseline across all benchmarks: +8.1% on LongVideoBench, +9.0% on LVBench, +4.2% on Video-MME Long, and +3.5% on MLVU Test. Notably, these gains are achieved while using 30-40% fewer frames on average, demonstrating that our adaptive sampling strategy effectively identifies and retrieves the most relevant visual information. VideoBrain also outperforms Video-R1, a recent reasoning-enhanced model, on most benchmarks despite using fewer frames. Compared to FrameThinker (He et al., 2025), a concurrent agentic video reasoning method, VideoBrain achieves higher accuracy (+4.7% on LVBench, +2.2% on Video-MME Long) while using fewer frames. We attribute this to two key differences: (1) FrameThinker relies solely on uniform temporal sampling, whereas our dual-agent design enables both semantic retrieval and temporal sampling; (2) our behavior-aware reward function explicitly discourages unnecessary agent invocations, leading to more efficient frame usage.

## 4.3. Ablation Studies

We conduct comprehensive ablation studies on LVBench to analyze the contribution of each component. Results are shown in Table 3.

**Agent Ablations.** Removing the CLIP Agent (w/o CLIP Agent) causes a significant drop from 41.3% to 36.3%, particularly affecting Entity (-6.7%) and Temporal (-16.6%) categories that require locating specific visual content. This validates the importance of semantic retrieval for questions where target information appears at unpredictable locations. Removing the Uniform Agent (w/o Uniform Agent) reduces accuracy to 39.8%, with notable drops in Event (-2.8%) and Temporal (-11.8%) categories that require understanding sequential actions within specific intervals. This confirms that temporally dense sampling complements semantic retrieval for comprehensive video understanding.

**Training Ablations.** Removing RL (w/o RL) reduces overall accuracy from 41.3% to 39.1%, showing that reinforcement learning with behavior-aware rewards helps refine the agent usage policy. Removing SFT (w/o SFT) leads to a larger drop to 35.5%, indicating that cold-start training is crucial for learning basic reasoning and tool usage abilities. Removing both (w/o SFT & RL) further degrades performance to 34.5%, only marginally better than the baseline (32.3%).

**Dataset and Reward Ablations.** Using unclassified SFT data (w/o SFT Classified Dataset) drops accuracy to 36.2%, demonstrating that our Direct/Adaptive/Active classification

*Table 3.* Ablation study on LVBench. We report accuracy (%) for each category and overall, along with average frames used.

| Model | Entity | Event | Key Info. | Reasoning | Summ. | Temporal | Overall | Frames |
|---|---|---|---|---|---|---|---|---|
| Qwen3-VL-8B-Instruct | 32.5 | 30.6 | 42.6 | 29.9 | 25.5 | 24.6 | 32.3 | 32.0 |
| **VideoBrain** | **42.0** | **38.7** | **50.0** | 34.7 | 38.2 | 47.7 | **41.3** | 21.6 |
| *Agent Ablations* | | | | | | | | |
| w/o CLIP Agent | 35.3 | 38.1 | 36.5 | 33.1 | **43.4** | 31.1 | 36.3 | 21.7 |
| w/o Uniform Agent | 41.5 | 35.9 | 48.9 | 38.5 | 35.8 | 35.9 | 39.8 | 22.8 |
| *Training Ablations* | | | | | | | | |
| w/o RL | 40.9 | 36.4 | 45.6 | 35.4 | 29.1 | **51.6** | 39.1 | 22.9 |
| w/o SFT | 37.4 | 32.8 | 41.7 | 30.1 | 40.0 | 30.8 | 35.5 | 23.4 |
| w/o SFT &RL | 34.8 | 32.6 | 43.6 | 31.5 | 40.0 | 21.9 | 34.5 | 25.0 |
| *Dataset and Reward Ablations* | | | | | | | | |
| w/o SFT Classified Dataset | 37.8 | 33.2 | 36.4 | 32.8 | 24.1 | 34.5 | 36.2 | 19.3 |
| w/o RL Classified Dataset | 40.0 | 38.9 | 51.1 | 38.2 | 27.8 | 50.0 | 40.8 | **18.8** |
| w/o Reward Shaping | 38.8 | 36.7 | 43.3 | 30.2 | 32.1 | 41.7 | 39.0 | 23.3 |
| w/o Reward Format Check | 40.6 | 35.7 | 48.4 | **39.0** | 38.9 | 41.5 | 39.7 | 23.3 |

helps create more effective training signals. Similarly, using randomly selected RL data (w/o RL Classified Dataset) reduces accuracy to 40.8%, showing that classified data also benefits RL training. Removing behavior-aware reward shaping (w/o Reward Shaping) reduces accuracy to 39.0%, confirming that category-specific rewards prevent reward hacking and encourage appropriate agent usage. Removing format checking (w/o Reward Format Check) slightly decreases performance to 39.7%, validating the importance of enforcing structured outputs.

### 4.4. Generalization Experiment

**Short Video Generalization.** To evaluate whether Video-Brain generalizes beyond long videos, we test on short video benchmarks without any fine-tuning. **DREAM-1K** (Wang et al., 2024a) is a video description benchmark featuring rich events, actions, and motions, with videos from 1 to 49 seconds. **Video-MME Short** (Fu et al., 2025) contains videos ranging from 11 seconds to 131 seconds.

As shown in Table 4, VideoBrain achieves +5.1% improvement on DREAM-1K and +2.4% on Video-MME Short compared to the Qwen3-VL-8B-Instruct baseline, while using 46-47% fewer frames. The significant reduction in frame usage demonstrates that VideoBrain automatically adapts its strategy when initial frames provide sufficient information, reducing agent invocation frequency on shorter videos. Furthermore, the strong performance on DREAM-1K validates that our approach enhances not only QA capabilities but also detailed video description generation.

*Table 4.* Generalization to short video benchmarks.

| Benchmarks | DREAM-1K | | Video-MME Short | |
|---|---|---|---|---|
| *Duration* | *1s - 49s* | | *11s - 131s* | |
| Model | Prec (%) | Frame | Acc (%) | Frame |
| Qwen3-VL-8B | 33.6 | 32.0 | 69.2 | 32.0 |
| **VideoBrain (Ours)** | **38.7** | **16.9** | **71.6** | **17.2** |
| Δ | +5.1 | -47% | +2.4 | -46% |

## 5. Conclusion

We presented VideoBrain, an end-to-end framework that enables Vision-Language Models to adaptively sample video frames through learned policies. Our approach addresses key limitations of prior work through three innovations: (1) dual sampling agents that combine semantic retrieval via CLIP with uniform temporal sampling, allowing flexible information gathering based on question type; (2) a behavior-aware reward function that prevents reward hacking by encouraging efficiency on questions answerable from initial frames while promoting exploration on those requiring additional visual information; and (3) an end-to-end architecture where the VLM directly perceives frames and makes sampling decisions, eliminating the information bottleneck of pipeline approaches. Experiments on four long video benchmarks demonstrate that VideoBrain achieves consistent improvements over the baseline (+3.5% to +9.0%) while using 30-40% fewer frames, and generalizes effectively to short video benchmarks without fine-tuning.

## Impact Statement

This paper presents work whose goal is to advance the field of Machine Learning. There are many potential societal consequences of our work, none of which we feel must be specifically highlighted here.

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

# A. System Prompt

Figure 3 shows the complete system prompt used during inference. The prompt defines two sampling tools (`uniform_sample` and `clip_sample`) and provides instructions for when to use each tool.

```
You are a helpful assistant to answer video questions.
You will initially see 16 uniformly sampled frames from the video.  These initial
frames may not be sufficient to answer the question accurately.  You can use the
provided tools to gather more targeted frames before answering.
# Tools
You can call one or more functions to assist with the user query, especially when
the initial frames are insufficient.  You are provided with function signatures
within <tools></tools> XML tags:
<tools>
[
  {
    ''type'':  ''function'',
    ''function'':  {
      ''name'':  ''uniform_sample'',
      ''description'':  ''Uniformly sample 8 frames between start_frame and
end_frame.'',
      ''parameters'':  { ... }
    }
  },
  {
    ''type'':  ''function'',
    ''function'':  {
      ''name'':  ''clip_sample'',
      ''description'':  ''Sample 4 frames within the frame range based on semantic
relevance to a text prompt.'',
      ''parameters'':  { ... }
    }
  }
]
</tools>
# How to call a tool
Return a JSON object with the function name and arguments inside XML tags:
<tool_call>
{''name'':  ''<function-name>'', ''arguments'':  { /* params */ }}
</tool_call>
# Instructions
1.  When you think there is insufficient information, you can obtain more frames by
calling uniform_sample or clip_sample.
2.  For problems such as inference, sorting, or summarization, uniform_sample can be
considered first.
3.  For visual positioning and confirming specific information, clip_sample can be
considered as a priority.
4.  When using clip_sample, a concise and to-the-point prompt is crucial.
```

*Figure 3.* System prompt for VideoBrain inference. The agent is instructed to use `uniform_sample` for temporal understanding and `clip_sample` for semantic retrieval.

Figure 4 shows the user prompt template for the initial turn, and Figure 5 shows the prompt template for subsequent turns after agent invocation.

# B. Training Curves

Figure 6 shows the training reward curves during the RL stage, demonstrating the effectiveness of our dual-agent design and SFT warm-start strategy.

```
Question: {question}
# Video Information:
- Total frames: {total_frames}
- FPS: {fps}
# Output Format:
Start with <thinking>. Format strictly as:
<thinking>...</thinking><tool_call>...</tool_call>
or
<thinking>...</thinking><answer>...</answer>
frame {idx_1}: <image>
frame {idx_2}: <image>
...
frame {idx_n}: <image>
```

*Figure 4.* User prompt template for the initial turn.

```
Start with <thinking>. Format strictly as:
<thinking>...</thinking><tool_call>...</tool_call>
or
<thinking>...</thinking><answer>...</answer>
Based on sampling, here are the following frames:
frame {idx_1}: <image>
frame {idx_2}: <image>
...
frame {idx_n}: <image>
```

*Figure 5.* Turn prompt template for subsequent turns after agent invocation. The sampled frames are appended with their frame indices.

**Agent Design Ablation (Figure 6a).** The left panel compares three variants: our full model with both agents (Ours, blue), a variant without the CLIP Sample Agent (w/o CLIP, red), and a variant without the Uniform Sample Agent (w/o uniform, green). All models show relatively stable training, but achieve different performance levels. The full model (Ours) reaches the highest average reward of 0.7480, significantly outperforming w/o CLIP (0.5998, $-24.71\%$) and w/o uniform (0.7116, $-5.12\%$). The larger gap when removing CLIP suggests that semantic retrieval contributes more to overall performance than uniform temporal sampling, which aligns with our intuition that locating relevant content across long videos is more challenging than densifying already-identified regions. Nevertheless, both agents are essential for comprehensive video understanding.

**SFT Warm-Start Impact (Figure 6b).** The right panel demonstrates the benefit of supervised fine-tuning for cold-start initialization. The model with SFT (w/ SFT, blue) begins training at a higher reward level (first 20 steps: 0.0433) and converges to 0.0448, exhibiting lower variance (std: 0.0014) throughout training. In contrast, the model without SFT (w/o SFT, red) starts from a lower baseline (first 20 steps: 0.0353) and requires approximately 50 steps to catch up, eventually reaching 0.0442 with higher variance (std: 0.0033). This 7.64% improvement validates our two-stage training strategy: SFT teaches basic reasoning and tool usage capabilities, providing a stable foundation for RL to further refine the agent invocation policy.

## C. Reward Design Detail

We provide implementation details for the reward function introduced in Section 3.4.

### C.1. Format Validation

The format gate $\mathbb{I}_{\text{format}}$ enforces structural integrity through the following validation rules:

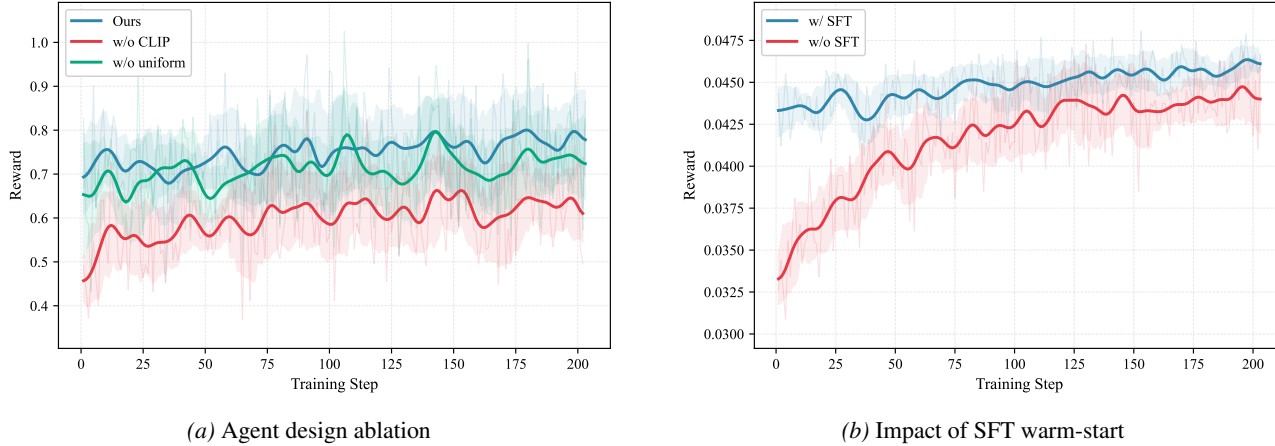

*(a)* Agent design ablation           *(b)* Impact of SFT warm-start

*Figure 6.* Training reward curves during reinforcement learning stage. **Left:** Comparison of our full model (Ours) against ablated versions removing either the CLIP Sample Agent (w/o CLIP) or the Uniform Sample Agent (w/o uniform). The full model achieves the highest reward (0.7480), outperforming w/o CLIP (0.5998) by 24.71% and w/o uniform (0.7116) by 5.12%, validating the complementary nature of dual agents. **Right:** Comparison of models with SFT warm-start (w/ SFT) versus without (w/o SFT). The SFT model starts at a higher baseline (0.0433) and converges to 0.0448, while the non-SFT model starts lower (0.0353) and reaches 0.0416, demonstrating a 7.64% improvement from supervised fine-tuning initialization.

**Tag Matching.** We verify that:

- the number of `<thinking>` equals `</thinking>`

- the number of `<tool_call>` equals `</tool_call>`

- the number of `<answer>` equals `</answer>`

- $\#\{\texttt{<thinking>}\} = \#\{\texttt{<tool\_call>}\} + \#\{\texttt{<answer>}\}$.

**Non-Empty Content.** All tag pairs must have non-empty content. We check that content within `<thinking>...</thinking>`, `<answer>...</answer>`, and `<tool_call>...</tool_call>` is not empty after stripping whitespace.

**Tool Call Validation.** Each `<tool_call>` must contain valid JSON. For `uniform_sample`, the format is:

```
{"name": "uniform_sample",
 "arguments": {"start_frame": <int>,
               "end_frame": <int>}}
```

For `clip_sample`, the format additionally requires a non-empty prompt:

```
{"name": "clip_sample",
 "arguments": {"start_frame": <int>,
               "end_frame": <int>,
               "prompt": <string>}}
```

We enforce start_frame $<$ end_frame for both tools.

**Duplicate Detection.** To prevent redundant exploration, we reject trajectories with duplicate tool calls:

- For `uniform_sample`: a call is considered duplicate if its frame range exactly matches or is within $\pm 1$ of any previous call.

- For `clip_sample`: a call is considered duplicate if its prompt exactly matches any previous call.

## C.2. Behavior-Aware Reward Details

Table 5 shows the complete behavior-aware reward matrix, including the partial reward for Active questions when the model uses agents but answers incorrectly.

*Table 5.* Complete behavior-aware reward $R_{\text{behavior}}$ matrix. The partial reward (+0.2) for Active questions encourages exploration even when the answer is incorrect, addressing reward sparsity for difficult questions.

| Category | Correct Answer | | Incorrect Answer | |
|---|---|---|---|---|
| | w/o Agent | w/ Agent | w/o Agent | w/ Agent |
| Direct | +0.5 | 0 | 0 | 0 |
| Adaptive | +0.5 | +0.5 | 0 | 0 |
| Active | 0 | +0.5 | 0 | +0.2 |

## C.3. LLM-as-Judge Scoring

For accuracy scoring $R_{\text{accuracy}}$, we use different LLM judges based on question type:

- **Multiple-choice (MC):** We use Qwen-Flash for strict binary scoring. The score is 1.0 if the model's answer matches the ground truth option, and 0.0 otherwise.

- **Open-ended (OE):** We use DeepSeek-V3.2 to assess semantic similarity between the model's answer and the ground truth, yielding continuous scores in $[0, 1]$.

## C.4. Some Discussion

### C.4.1. WHY NOT DESIGN A REWARD SYSTEM SPECIFICALLY FOR EACH TYPE OF AGENT'S ACTIONS?

Since our dual-agent system operates through mutual collaboration rather than independent actions, designing agent-specific reward systems poses significant challenges. While our results show that many questions can be answered correctly with a single agent turn, difficult cases benefit substantially from collaborative multi-agent interactions (see Figure 7 for an example).

However, determining which questions require collaborative agent assistance versus single-agent solutions would demand prohibitively high annotation costs. This task requires extensive human effort and exhaustive video testing to properly label the optimal agent strategy for each case.

VideoBrain's design philosophy prioritizes providing an easily implementable data pipeline to enhance video understanding capabilities. Rather than introducing complex, task-specific designs that would complicate the data annotation process, we opt for a unified reward framework that naturally encourages effective agent collaboration when needed, while maintaining simplicity and scalability in our training pipeline.

### C.4.2. WHY ARE FORMATTING CHECKS NECESSARY?

In our reward design, if the model's output fails format validation, we do not evaluate answer accuracy and directly assign a score of zero. This strict policy is intentional: even if the model may have produced a correct answer, malformed output constitutes noise that should not be rewarded.

Despite supervised fine-tuning for cold-start initialization, the model still exhibits unreasonable behaviors when invoking agents. Common issues include generating extra `<tool_call>` tags, producing empty content between tags, or outputting malformed JSON. We therefore design a comprehensive validation protocol to penalize such behaviors.

This is particularly important for preventing agent abuse. We observe that when format checking is disabled, the model tends to output numerous duplicate or redundant tool calls during evaluation. This behavior represents another form of noise: if the model happens to answer correctly despite the chaotic output, it would receive undeserved rewards, thereby corrupting the training signal. Our strict format gate ensures that only well-structured reasoning trajectories contribute to policy optimization.

## D. Error Analysis

We analyze failure cases where VideoBrain invokes agents but still fails to answer correctly. We manually annotated 70 of such cases and categorized each by error type, as shown in Table 6.

*Table 6.* Failure case categorization by error type.

| Error Type | Ratio % |
|---|---|
| Imprecise Agent Sampling | 54.3 |
| Visual Hallucination | 20.0 |
| Incomplete Exploration | 17.1 |
| Failed Answer Grounding | 5.7 |
| Wrong Tool Selection | 2.9 |

**Imprecise Agent Sampling** accounts for the majority of failures and stems from two sub-causes: (1) *incorrect frame range specification*: the model selects an interval that is too wide or inadvertently excludes the answer frame; (2) *imprecise CLIP prompt*: the text query does not accurately capture the visual semantics. For example, for the question *"What is the first warning on the warning sign in the volcano skali?"*, the model issued a CLIP query of ``volcano Skali warning sign lava``, whereas a more precise query such as ``warning notice`` would have retrieved the relevant frame more effectively. Overly verbose prompts dilute the semantic signal and reduce retrieval precision.

**Incomplete Exploration** (17.1%) reflects cases where the model terminates reasoning too early. While 62.6% of questions are answered in round 1, some harder questions require additional agent calls to achieve meaningful accuracy improvement, suggesting that the model's difficulty awareness is not yet perfect. We regard both error types as directions for future work, including better agent prompt generation and more adaptive round-allocation strategies.

## E. Computational Cost and Latency

We use SigLIP2-ViT-B/16 as the CLIP encoder, which is lightweight. Processing 256 frames requires 5 TFLOPs, introducing negligible computational overhead, as summarized in Table 7.

*Table 7.* SigLIP2 encoder computational cost and model size.

| Method | FLOPs | Avg Running Time[†] | Weights |
|---|---|---|---|
| SigLIP2 (256f) | 5 TFLOPs | 3.1s | 0.3 GB (FP32) |

[†] Measured concurrently with VideoBrain inference on the same GPU, reflecting the actual deployment scenario.

Table 8 presents end-to-end latency on LVBench, measured on a single H20 GPU. To ensure a fair comparison, the CLIP encoder and VideoBrain share the same GPU during inference, accounting for the additional memory overhead introduced by CLIP. VideoBrain outperforms 256-frame inference in both accuracy (41.3% vs. 40.4%) and speed (30.5s vs. 90.4s). The higher latency relative to the 32-frame baseline is due to reasoning mode and multi-turn interaction.

*Table 8.* End-to-end latency on LVBench (single H20 GPU).

| Method | Acc (%) | Frames | Latency |
|---|---|---|---|
| Qwen3-VL-8B | 32.3 | 32 | 5.1s |
| Qwen3-VL-8B | 40.4 | 256 | 90.4s |
| VideoBrain | 41.3 | 21.6 | 30.5s |

## F. VideoBrain Case Study

This section illustrates how VideoBrain selectively invokes different agents based on question characteristics through representative examples.

## F.1. Multi Agent Collaboration Case

Figure 7 demonstrates how VideoBrain coordinates both agents to answer complex questions in extremely long videos. Given the question "What color of the jacket does Sophia wear when she sees the crane status?" in a video with over 209,000 frames, the model must locate both the character Sophia and the specific moment involving cranes.

In the first iteration, VideoBrain uses the CLIP Sample Agent with the prompt "crane status" to semantically search the entire video, identifying crane-related frames around 154434-165114. In the second iteration, recognizing that Sophia is not visible in the returned frames, the model invokes the Uniform Sample Agent to densely sample the identified temporal region [154434, 165114]. In the third iteration, the model observes Sophia wearing a pink jacket in a car (frames 157485, 159010) near the crane scenes, correctly concluding the answer is "Pink."

In contrast, the baseline Qwen3-VL-8B-Instruct sees Sophia in multiple scenes wearing different colored jackets but cannot determine which moment corresponds to seeing the crane, answering "Unknown." This case highlights VideoBrain's ability to combine semantic retrieval (CLIP) with temporal densification (Uniform) through multi-turn collaboration, effectively solving questions that require locating specific events in extremely long videos.

## F.2. CLIP Sample Case

We present two cases demonstrating how VideoBrain leverages the CLIP Sample Agent for semantic retrieval in different scenarios.

**Finding Specific Visual Evidence.** Figure 9 shows how VideoBrain uses CLIP retrieval to locate specific visual content. Given the question "What mode of transportation does the male protagonist use to get home from work?", the model identifies frame 2871 with the text "19:45 - home" (in Russian) indicating the going-home scene, but cannot clearly see the transportation mode. VideoBrain then uses the CLIP Sample Agent with the prompt "going home transportation" within the narrowed range [2800, 3078]. The retrieved frames clearly show two yellow electric scooters, enabling the correct answer "Riding an electric scooter." The baseline sees only the man walking with a backpack and incorrectly concludes that none of the options match.

**Event Ordering with Sparse Activities.** Figure 8 demonstrates how VideoBrain uses iterative CLIP retrieval to answer event ordering questions. Given the question asking for the chronological order of four activities (water sliding, making jewelry, paragliding, playing harp), the model must locate all activities and determine their sequence. In the initial 16 frames, VideoBrain only identifies the harp scene (frame 11485) and recognizes the other three activities are missing. It then systematically uses the CLIP Sample Agent to retrieve frames for each activity: "water sliding" (frames 6443, 7234), "making jewelry" (frames 339, 452), and "paragliding" (frames 1243, 1356). With all activities located, the model determines the order by frame indices: making jewelry (339) $\rightarrow$ paragliding (1243) $\rightarrow$ water sliding (6443) $\rightarrow$ playing harp (11485), correctly answering option B. The baseline Qwen3-VL-8B-Instruct with 32 frames fails to capture all four activities and incorrectly answers "None of the options accurately represent the order."

## F.3. Uniform Sample Case

We present two cases demonstrating how VideoBrain leverages the Uniform Sample Agent for different reasoning scenarios.

**Narrowing Search Range Based on Visual Clues.** Figure 10 illustrates how VideoBrain progressively narrows down the search range through visual reasoning. Given the question "What did the big snake turn into after being defeated in the video?", the model analyzes the initial frames and identifies a snake-like creature with a crown in frame 7049. Recognizing that the defeat scene is not visible, the model reasons that "the defeat scene would likely be in the middle of the video" and invokes the Uniform Sample Agent with the range [5000, 10000]. Upon receiving additional frames, it identifies frame 9284 showing the snake with purple substance dripping, correctly concluding the answer is "Sewage." In contrast, although the baseline Qwen3-VL-8B-Instruct receives 32 uniformly sampled frames, it misinterprets the visual effect as "a shimmering, sparkling substance" and incorrectly answers "Clean water." This demonstrates VideoBrain's ability to leverage visual clues to intelligently narrow down the temporal search space and locate the critical evidence.

**Computing Frame Indices from Timestamp Hints.** Figure 11 demonstrates VideoBrain's capability to perform numerical reasoning for precise frame localization. Given the question "What is the maid doing from 13:11-13:24?" with a video

---

***Question***: What color of the jacket does Sophia wear when she sees the crane status?
A. Pink B. Black C. White D. Blue
***Ground Truth***: A. Pink

---

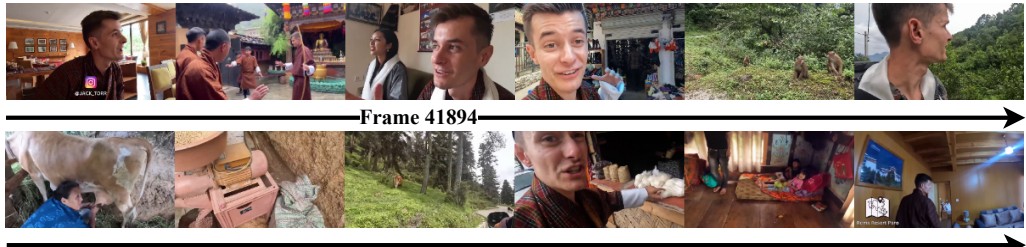

**Frame 41894**

---

***Qwen3-VL-8B-Instruct***: *<thinking>...I can see Sophia in several scenes. In one frame, she is seen wearing a pink jacket while sitting in a car. In another frame, she is wearing a brown jacket. However, there is no frame showing her seeing a crane status...Therefore, I cannot determine the color of her jacket at that moment. </thinking>*
*<answer>Unknown.</answer>*

---

***VideoBrain Iteration 1***: *<thinking>Looking at the frames: ...Frame 41894: Shows a man and a woman sitting together. The woman has a white scarf around her neck. This could be Sophia. ...Since the question is about a specific moment (when Sophia sees the crane status), and I don't see this information in the provided frames, ...Let me call clip_sample with a prompt about "crane status" to get relevant frames.</thinking>*
*<tool_call>{"name": "clip_sample", "arguments": {"start_frame": 0, "end_frame": 209474, "prompt": "crane status"}}</tool_call>*

---

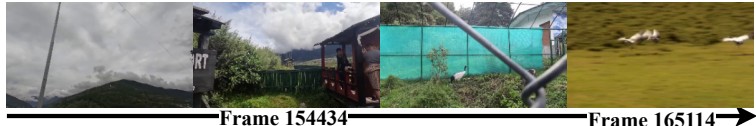

**Frame 154434**    **Frame 165114**

---

***VideoBrain Iteration 2***: *<thinking>The clip_sample returned frames: Frame 154434: Shows a scene with a sign that says "ORT" and some people near what appears to be a crane or similar structure. Frame 164292: Shows a crane (bird) in an enclosure with a green fence. Frame 165114: Shows cranes flying in a field. However, I don't see Sophia in any of these frames, ...The crane-related frames are around frames 154434-165114, so let me call uniform_sample with the range 154434-165114 to get more frames where Sophia might appear.</thinking>*
*<tool_call>{"name": "uniform_sample", "arguments": {"start_frame": 154434, "end_frame": 165114}}</tool_call>"*

---

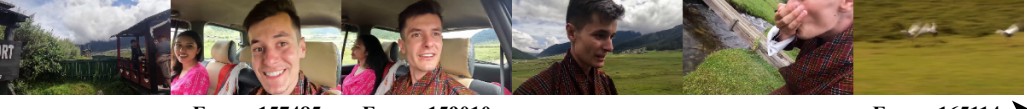

**Frame 157485**  **Frame 159010**      **Frame 165114**

---

***VideoBrain Iteration 3***: *<thinking>Looking at the frames from 154434 to 165114... I see that in frames 157485 and 159010, Sophia is in a car. It's possible that Sophia sees the cranes from there. The last frame (165114) shows cranes flying, but I don't see Sophia in it. However, Sophia is wearing a pink jacket in the car frames, and she's likely in the car when she sees the cranes, the answer should be A. Pink.</thinking>*
*<answer>A. Pink </answer>*

---

*Figure 7.* Multi-Agent Collaboration Case: VideoBrain first uses CLIP Sample to locate crane-related scenes, then uses Uniform Sample to densely sample the identified region, finding Sophia in a pink jacket.

*Question*: Please identify the option that corresponds to the order of events as they occur in the video
A. water sliding --> making jewelry --> paragliding --> playing harp
B. making jewelry --> paragliding --> water sliding --> playing harp
C. playing harp --> making jewelry --> water sliding --> paragliding
D. water sliding --> playing harp --> making jewelry --> paragliding
E. paragliding --> water sliding --> playing harp --> making jewelry
F. water sliding --> making jewelry --> playing harp --> paragliding
*Ground Truth*: B. making jewelry --> paragliding --> water sliding --> playing harp

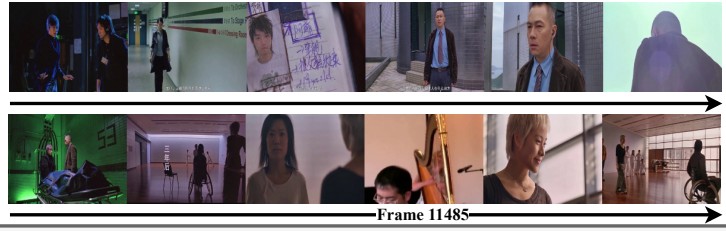

Frame 11485

*Qwen3-VL-8B-Instruct*: <thinking>The provided video frames do not depict any of the activities listed in the options (water sliding, paragliding  making jewelry, playing harp)... </thinking>
<answer>None of the options accurately represent the order of actions shown in the video.</answer>

*VideoBrain Iteration 1*: <thinking>...I need to identify which frames correspond to the four activities mentioned in the options...The harp scene (frame 11485) is clearly visible. The other activities mentioned in the options (water sliding, making jewelry, paragliding) don't appear to be represented in the provided frames... I'll use the clip_sample tool to get frames that might contain the missing activities. </thinking>
<tool_call>{"name": "clip_sample", "arguments": {"start_frame": 0, "end_frame": 14357, "prompt": "water sliding"}}</tool_call>

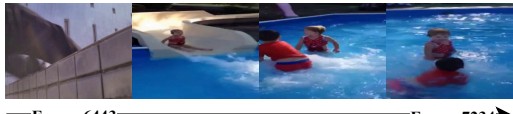

Frame 6443                    Frame 7234

*VideoBrain Iteration 2*: <thinking>...The user has now provided additional frames from the clip_sample tool for "water sliding": Frame 6443: Shows what appears to be a water slide Frame 7008: Shows someone on a water slide Frame 7121: Shows children in a pool Frame 7234: Shows children in a pool. This confirms that water sliding is one of the activities in the video... Since I don't have enough information about the order of all four activities, I should use the clip_sample tool again to find frames that might show making jewelry and paragliding.</thinking>
<tool_call>{"name": "clip_sample", "arguments": {"start_frame": 0, "end_frame": 14357, "prompt": "making jewelry"}}</tool_call>

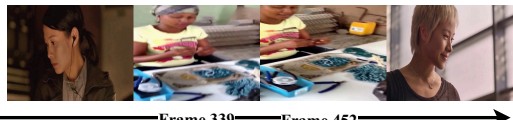

Frame 339        Frame 452

*VideoBrain Iteration 3*: <thinking>...The user has provided additional frames from the clip_sample tool for "making jewelry": Frame 339 and Frame 452: Shows someone making jewelry (hands working with materials). This confirms that making jewelry is one of the activities... I still need to find evidence of paragliding. I'll use the clip_sample tool to find frames that might show paragliding.</thinking>
<tool_call>{"name": "clip_sample", "arguments": {"start_frame": 0, "end_frame": 14357, "prompt": "paragliding"}}</tool_call>

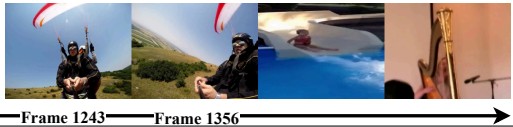

Frame 1243        Frame 1356

*VideoBrain Iteration 4*: <thinking>...The user has now provided additional frames from the clip_sample tool for "paragliding": Frame 1243 and Frame 1356: Shows someone paragliding (clearly visible). I need to determine the order of these activities... The frame numbers suggest that making jewelry (339, 452) comes before water sliding (7008), which comes before paragliding (1243, 1356), which comes before playing harp (11485). So the order would be: making jewelry → water sliding → paragliding → playing harp, which matches option B.</thinking>
<answer>B. making jewelry --> paragliding --> water sliding --> playing harp </answer>

*Figure 8.* CLIP Sample Case (Event Ordering): VideoBrain iteratively retrieves frames for four activities using semantic search, then determines their chronological order by frame indices.

> ***Question****: What mode of transportation does the male protagonist use to get home from work in the video?*
> *A. Riding an electric scooter. B. Driving a car. C. Cycling. D. Riding an electric bike.*
> ***Ground Truth****: A. Riding an electric scooter*

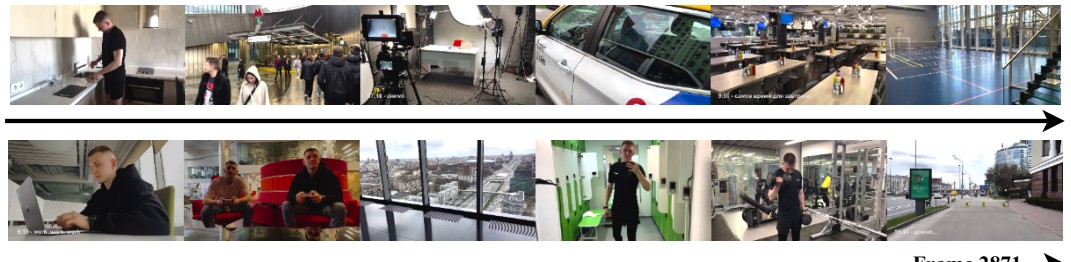

**Frame 2871**

> ***Qwen3-VL-8B-Instruct****: <thinking>...At 19:45, a text overlay appears that says "домой", which translates to*
> *"home." At 19:30, shows the man walking out of a building with a backpack, and the text says "19:30 - домой" meaning*
> *"19:30 - home". ...The only mode of transportation shown is walking. ...none of the options are correct...</thinking>*
> *<answer>None of the options accurately represent the order of actions shown in the video.</answer>*

> ***VideoBrain Iteration 1****: <thinking>Looking at the frames: Frame 2871 shows a street scene with the text "19:45 -*
> *домой..." which translates to "19:45 - home...". This suggests this is the scene where he's going home. However, I don't*
> *see a clear indication of what mode of transportation he's using in this frame. ...I'll use clip_sample to find frames*
> *related to "going home" or "transportation" around the 19:45 time frame.</thinking>*
> *<tool_call>{"name": "clip_sample", "arguments": {"start_frame": 2800, "end_frame": 3078,*
> *"prompt": "going home transportation"}}</tool_call>*

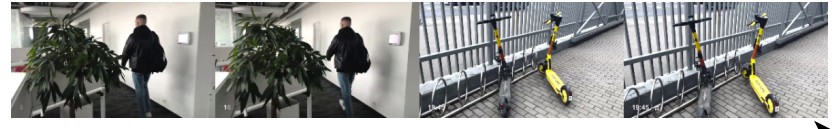

> ***VideoBrain Iteration 2****: <thinking>...The key frame provided by the clip_sample tool shows two electric scooters*
> *parked at 19:45, which matches the time mentioned in the question. The frame also shows the text "19:45 - домой"*
> *(19:45 - home) which indicates this is the scene where he's going home. The frame doesn't show him driving a car,*
> *cycling, or riding an electric bike... The correct answer should be A. Riding an electric scooter.</thinking>*
> *<answer>A. Riding an electric scooter</answer>*

*Figure 9.* CLIP Sample Case (Visual Evidence): VideoBrain uses targeted semantic retrieval to find transportation-related frames within a localized time range.

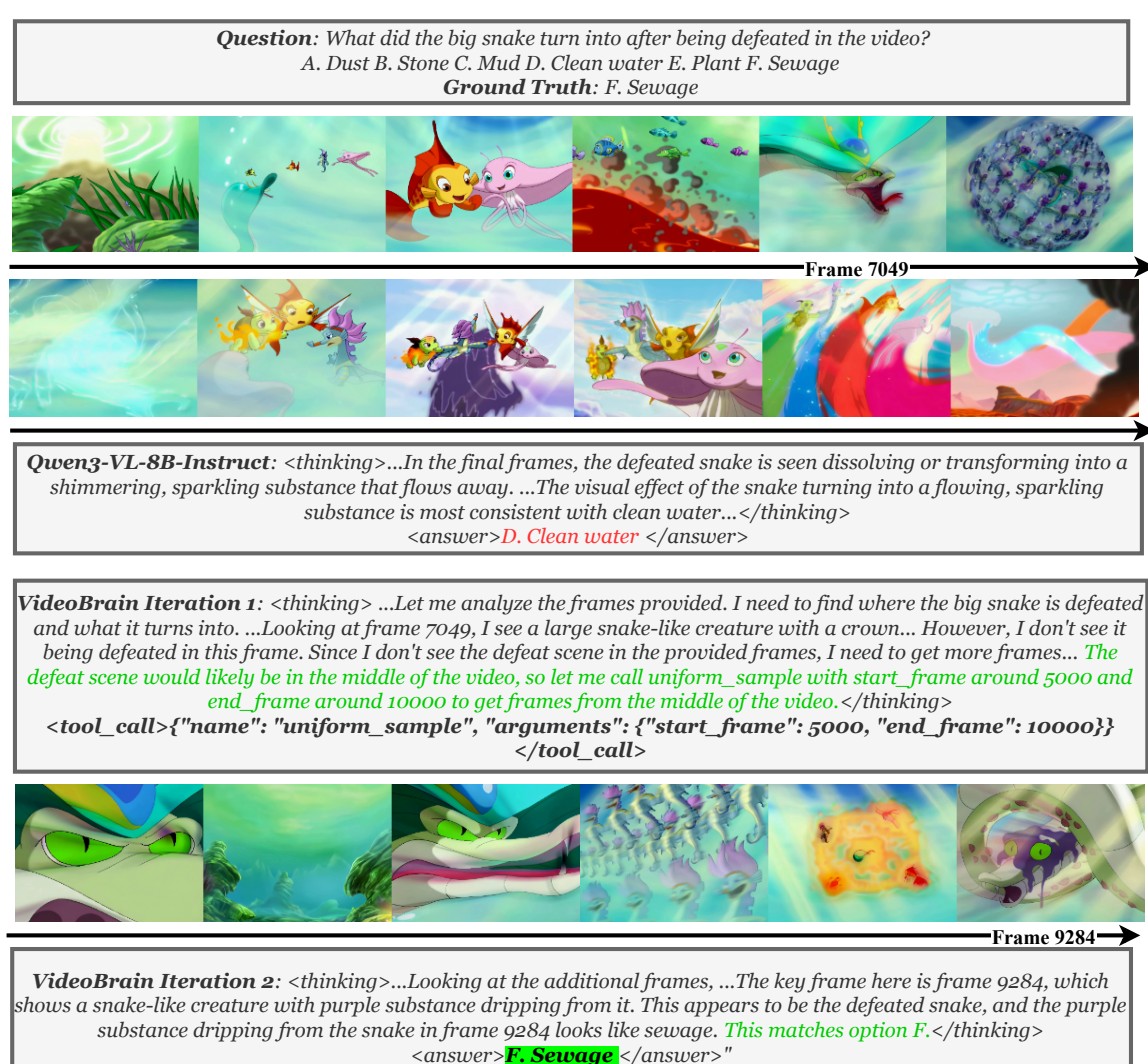

*Figure 10.* Uniform Sample Case: Narrowing search range based on visual clues. The model reasons about the snake's location and samples the middle portion of the video to find the defeat scene.

at 23.97 FPS, the model computes the frame indices during reasoning: $13 : 11 = 791 \sec \times 23.97 \approx 18,947$ frames and $13 : 24 = 804 \sec \times 23.97 \approx 19,277$ frames. The model then invokes the Uniform Sample Agent with the calculated range [18947, 19277] and correctly identifies that the maid is skipping rope. In contrast, the baseline Qwen3-VL-8B-Instruct fails as the initial frames do not cover this timestamp range. This highlights VideoBrain's ability to combine numerical reasoning with adaptive sampling for timestamp-based queries.

**Question**: *What is the maid doing from 13:11-13:24?*
*A. She is skipping rope B. She is kicking shuttlecock C. She is weaving a flower basket D. She is weaving clothes*
**Ground Truth**: *A. She is skipping rope*

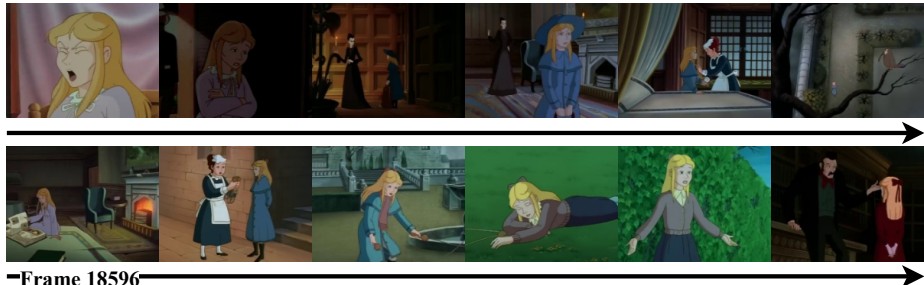

**Frame 18596**

**Qwen3-VL-8B-Instruct**: *<thinking> The provided video frames do not contain any scene or action that depicts a maid performing any of the activities listed in options A, B, C, or D between 13:11-13:24... </thinking>*
*<answer>**None of the above**</answer>*

**VideoBrain Iteration 1**: *<thinking>...The video has a frame rate of 23.97 FPS. 13:11 = 13 minutes and 11 seconds = 791 seconds 13:24 = 13 minutes and 24 seconds = 804 seconds. Now, convert these to frames: Frame at 13:11 = 791 \* 23.97 ≈ 18,947 frames. Frame at 13:24 = 804 \* 23.97 ≈ 19,277 frames. So I need to look at frames between approximately 18,947 and 19,277. The provided video frames do not contain any scene or action that depicts a maid performing any of the activities listed in options A, B, C, or D between 13:11-13:24.... I should use the uniform_sample tool to get frames in that range. </thinking>*
***<tool_call>{"name": "uniform_sample", "arguments": {"start_frame": 18947, "end_frame": 19277}}***
***</tool_call>***

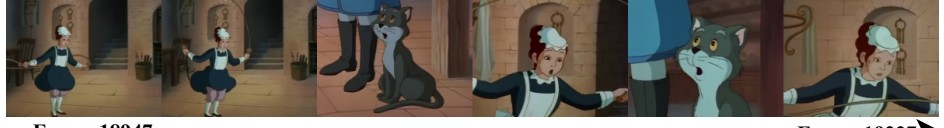

**Frame 18947** **Frame 19227**

**VideoBrain Iteration 2**: *<thinking>Looking at the provided frames from the uniform_sample call, ...I can see that the maid is skipping rope. In multiple frames, she is shown holding and using a rope for skipping. Let me check the options: A. She is skipping rope - This matches what I see in the sampled frames.</thinking>*
*<answer>***A. She is skipping rope***</answer>*

*Figure 11.* Uniform Sample Case: Computing frame indices from timestamp hints. The model converts timestamps to frame numbers and precisely samples the target temporal range.

