# OpenReview forum: "VideoBrain: Learning Adaptive Frame Sampling for Long Video Understanding"
_ICML.cc/2026/Conference — ICML 2026 regular_

### Official Review · Reviewer_dhbh · 2026-03-12

**Soundness:** 3
**Presentation:** 3
**Significance:** 3
**Originality:** 3
**Overall Recommendation:** 5
**Confidence:** 4

**Summary:**

This paper introduces VideoBrain, an e2e VLM agent framework. Different from vanilla agent RL methods, VideoBrain uses a novel behavior-aware reward to address the reward hacking issue. Experiment results on four long video understanding benchmarks show that VideoBrain is more accurate with less computation.

**Compliance With Llm Reviewing Policy:**

Affirmed.

**Final Justification:**

Thanks for the insightful clarification and additional experiments. My concern has been fully addressed. I decided to raise my score from 4 to 5.

**Key Questions For Authors:**

Please refer to Weaknesses.

Additional Questions:

1. Since SFT data is generated by Qwen3-VL-235B, what is its zero-shot performance?

**Limitations:**

yes

**Strengths And Weaknesses:**

Strengths.

1. This paper is well written and easy to understand. The method is intuitive.
2. The improvement across four long video understanding benchmarks is significant. Compared with the previous agentic framework, FrameThinker, there is consistent improvement.
3. Ablation studies on agents, training, dataset, and reward are provided.

Weaknesses

1. The contribution of the behavior reward is limited. First, the reward hacking problem is not well explained. In practice, most advanced RL algorithms (like DAPO) penalize long responses, which means the models have already been “discouraged” from invoking. Second, empirically, FrameThinker (w/o behav loss) and VideoBrain (with behav loss) use almost the same number of frames. This suggests that the behavior loss may not be effective.
2. Although many previous works also miss it, reporting the computation cost of the clip sample agent is highly recommended. The reason is that, for the most recent VLMs (e.g., Qwen3-VL), performance with sufficient frames is much better (e.g., on LVBench: Qwen3-VL-8B 32.3@32f vs 58.0@640f). Since computing clip features for all frames (used in the clip sample agent) is extremely expensive, it is necessary to make all parts of the computation transparent to support the overall performance–computation trade-off.

---

> ### Author Rebuttal · Authors · 2026-03-31
>
> We thank the reviewer for the detailed feedback. We address each concern below.
>
> **[W1] Contribution of the Behavior-Aware Reward.**
>
> We clarify two points.
>
> **On the comparison with length penalty (e.g., DAPO).** In prior agentic methods, the model receives a bonus whenever it invokes an agent *and* answers correctly. This creates a perverse incentive: since calling an agent and getting the right answer always yields more reward than answering directly, the model learns to invoke agents indiscriminately across all questions to maximize cumulative bonus — even on questions that can be answered from the initial frames alone. A length penalty (as in DAPO) partially counteracts this, but as long as the per-invocation bonus can outweigh the penalty (which is often the case for harder questions), indiscriminate invocation remains the optimal strategy. Our behavior-aware reward resolves this by being **question-type-aware**: agent invocation is rewarded only for Active questions (where additional frames are genuinely needed) and penalized for Direct questions (where initial frames suffice), removing the incentive to invoke agents unnecessarily.
>
> **On the similar frame counts between FrameThinker and VideoBrain.** This observation does not imply that behavior reward is ineffective. The goal of behavior reward is not simply to reduce frame count, but to ensure agents are invoked at the **right time**. With 16 initial frames, our model learns that **62.6% of long-video questions can be answered directly** without further agent calls (based on statistics over 2,298 samples). The ablation confirms this: removing reward shaping (w/o Reward Shaping) causes accuracy to drop from 41.3% to 39.0% ($-$2.3%) while frame usage increases from 21.6 to 23.3, showing that the model invokes agents more frequently but less effectively without behavior-aware supervision.
>
> **[W2] Computational Cost of the CLIP Agent.**
> We first clarify the frame count behind the 58.0 LVBench result. According to the Qwen3-VL Team: *"For Qwen3-VL, we trained the model with a context length of 256K, enabling it to process significantly more frames. Accordingly, our evaluations on LVBench take at most 2,048 frames (224K tokens, i.e., total\_pixels=224 * 1024 * 32 * 32)."* Therefore, the 58.0 result is achieved with up to 2,048 frames (224K tokens), not 640 frames. Processing 224K tokens requires substantial GPU memory.
>
> We use SigLIP2-ViT-B/16 as the CLIP encoder, which is lightweight. Processing 256 frames requires $\approx$5 TFLOPs, introducing small computational overhead.
>
> | Method | FLOPs | Avg Running Time† | Weights |
> |--------|-------|-----------------|---------|
> | SigLIP2 (256f) | $\approx$5 TFLOPs | 3.1s | $\approx$0.3 GB (FP32) |
>
> † Measured concurrently with VideoBrain inference on the same GPU, reflecting the actual deployment scenario.
>
> The following table presents End-to-end latency on LVBench, measured on a single H20 GPU. To ensure a fair comparison, the CLIP encoder and VideoBrain share the same GPU during inference, accounting for the additional memory overhead introduced by CLIP. VideoBrain outperforms 256-frame inference in both accuracy (**41.3%** vs. 40.4%) and speed (**30.5s** vs. 90.4s). The higher latency vs. the 32-frame baseline is due to reasoning mode and multi-turn interaction.
>
> | Method | Acc (%) | Frames | Latency |
> |--------|---------|--------|---------|
> | Qwen3-VL-8B (32f) | 32.3 | 32 | 5.1s |
> | Qwen3-VL-8B (256f) | 40.4 | 256 | 90.4s |
> | **VideoBrain** | **41.3** | **21.6** | **30.5s** |
>
> **[Q1] Zero-Shot Performance of Qwen3-VL-235B.**
>
> | Model | LVBench | Video-MME Long | LongVideoBench | MLVU Test | Frames |
> |-------|---------|----------------|----------------|-----------|--------|
> | Qwen3-VL-235B (zero-shot) | 47.0 | 63.2 | 66.6 | 57.96 | 32 |
> | Qwen3-VL-8B (zero-shot) | 32.3 | 45.6 | 45.2 | 43.4 | 32 |
> | **VideoBrain (8B)** | **41.3** | **49.8** | **53.3** | **46.9** | $\approx$21 |
>
> Qwen3-VL-235B serves as our teacher model for SFT data generation. We acknowledge that a performance gap remains between VideoBrain (8B) and the 235B teacher, which is expected given the large difference in model capacity. Nonetheless, VideoBrain significantly improves over the 8B baseline and partially closes the gap to the teacher, demonstrating that our training pipeline effectively transfers and refines knowledge beyond what the base model can achieve.

---

> > ### Author Rebuttal · Reviewer_dhbh · 2026-03-31
> >
> > W1 follow-up: The authors agree that a length penalty (as in DAPO) partially counteracts the invocation, but the weight may be not large enough for harder questions. Does this imply that increasing the conventional length penalty already solves the invocatio problem? So my follow-up question here is that, principally, what is the key novelty of your method from previous length penalty?
> >
> > W2 follow-up: Thanks for pointing out that the actual frame number is 2048. Just want to know some details. (1) Are you using qwen_vl_utils.process_vision_info to preprocess video inputs? Do you count this part into the latency? (2) Do you enable thinking when using Qwen3? I think Qwen3 enables thinking by default so that the generated tokens are much more than other methods. So I want to know if the major difference in latency is caused by pre-filling or just more reasoning tokens.

---

> > > ### Author Response · Authors · 2026-04-01
> > >
> > > We thank the reviewer for the follow-up questions and address each point below.
> > >
> > > **[W1 Follow-up] Key Novelty of Behavior-Aware Reward vs. Length Penalty.**
> > >
> > > Increasing the conventional length penalty does **not** solve the invocation problem, because the two mechanisms are essentially different.
> > >
> > > A length penalty (e.g., DAPO) applies a **length-based, content-agnostic** penalty to outputs, regardless of whether the length is caused by necessary or unnecessary agent calls. In a video understanding setting where some questions genuinely require additional frames (Active questions), a larger length penalty would indiscriminately suppress all agent invocations (including necessary ones), forcing the model to guess answers from insufficient visual information. A length penalty operates on output length, not on question semantics — it cannot inherently distinguish whether an agent invocation is necessary or not.
> > >
> > > Our behavior-aware reward is **question-type-aware** by design. Rather than penalizing output length, it directly shapes the agent invocation policy based on question category:
> > > - For **Direct** questions (initial frames suffice): the model is rewarded only when it answers correctly *without* invoking agents, actively discouraging unnecessary calls.
> > > - For **Active** questions (additional frames genuinely needed): the model is rewarded for invoking agents to gather additional information when initial frames are insufficient.
> > >
> > > This is an essentially different mechanism: length penalty penalizes output length regardless of invocation necessity, while behavior reward operates on whether the invocation behavior matches the actual information need of each question. The key novelty is the shift from length-based penalization to behavior-based guidance conditioned on question type.
> > >
> > > **[W2 Follow-up] Latency Details.**
> > >
> > > **(1) Preprocessing and latency accounting.**
> > > We do not use `qwen_vl_utils.process_vision_info`. Frame extraction and resizing are handled by our own implementation, and to ensure a fair comparison, the same procedure is applied consistently to both the baseline and VideoBrain. Our reported latency includes image preprocessing. Additionally, when the CLIP Agent is invoked, the time of sampling 128/256 candidate frames and CLIP inference is also included in the VideoBrain latency measurement.
> > >
> > > **(2) Thinking mode and latency breakdown.**
> > > We use **Qwen3-VL-8B-Instruct** as both the baseline and the backbone for VideoBrain. This model does not support thinking; the Qwen3-VL Team release provides Instruct and Thinking as two separate model weights, and thinking requires deploying Qwen3-VL-8B-Thinking separately. VideoBrain is fine-tuned on trajectories distilled from Qwen3-VL-235B-A22B-Thinking, and therefore learns to produce `<thinking>` reasoning chains during inference.
> > >
> > > The major latency difference of VideoBrain is caused by **reasoning tokens (decoding) and agent invocation, not visual token pre-filling**. VideoBrain uses on average only **21.6 frames**, fewer than even the 32-frame baseline, so its visual token pre-filling cost is actually lower than the baseline. The 30.5s latency comes primarily from the `<thinking>` reasoning process (average output length per sample $\approx$842 words across all turns vs. $\approx$5 words for the 32-frame baseline, measured on LVBench) and CLIP Agent calls ($\approx$3.1s per invocation). In contrast, the 256-frame baseline's 90.4s is dominated by visual token pre-filling (5.1s → 90.4s when scaling from 32 to 256 frames). Therefore, the latency overhead in VideoBrain is a direct consequence of its reasoning capability and agent invocation.

---

### Official Review · Reviewer_EXmM · 2026-03-12

**Soundness:** 4
**Presentation:** 4
**Significance:** 2
**Originality:** 3
**Overall Recommendation:** 4
**Confidence:** 5

**Summary:**

This paper proposes VideoBrain, designed to address the tension faced by MLLMs in long-video understanding between computational constraints and effective information extraction. Existing methods typically rely on uniform sampling, which easily misses critical information, or one-shot keyframe selection, which cannot correct initial selection errors. Although recent agent-based approaches enable iterative information gathering, they depend on text-only LLMs, creating an information bottleneck when visual content is converted into text. To overcome these limitations, VideoBrain allows the MLLMs to directly perceive video frames and iteratively reason about whether additional information is needed. The framework includes two sampling agents: a CLIP Sample Agent for semantic retrieval and a Uniform Sample Agent for local sampling. To prevent the model from abusing the agents to maximize rewards, the authors introduce a behavior-aware reward function based on dual-model data categorization into three types. Extensive experiments show that the method achieves substantial accuracy gains across multiple long-video benchmarks while reducing the number of required frames.

**Compliance With Llm Reviewing Policy:**

Affirmed.

**Key Questions For Authors:**

Please see the Weakness.

**Limitations:**

Limitations: No.
Societal impact: Yes, they discussed.

**Strengths And Weaknesses:**

### **Strengths**
- **The dual-agent design is novel.** Prior work mainly relies on iteratively dense sampling within specific intervals, whereas this approach combines CLIP-based agent with uniform sample agent. The CLIP agent locates semantically relevant visual content, while the uniform sample agent captures fine-grained temporal dynamics within selected segments. Together, they better address the information needs of VQA.
- **The reward design is effective.** Partitioning the training data into Direct, Adaptive, and Active categories helps the model learn when the available information is sufficient to answer a question, thereby reducing unnecessary exploration and reasoning.
- The paper is well written, and the proposed method is intuitive, effective, and easy to follow.

### **Weaknesses**
- **Main concern: the experimental setting is too constrained.** The paper evaluates long-video understanding with a very limited number of frames, which may overstate the advantage of the proposed agent method. With more frames, even a smaller model can achieve substantially better results, potentially reducing the claimed benefit of the method.

  | Model                    | Frames | LongVideoBench (1-60 min)          | Video-MME Long           |
  |--------------------------|--------|--------------------------|--------------------------|
  | Qwen3-VL 8B              | 32     | 45.2                     | 45.6                     |
  | VideoBrain (Qwen3-VL 8B) | ~22    | 53.3                     | 49.8                     |
  | Qwen3-VL 4B              | 2048   | 59.3 (reviewer-evaluated) | 61.3 (reviewer-evaluated) |

- **Suggested experiments.**
  1. Evaluate under higher-frame settings, e.g., whether VideoBrain can still help Qwen3-VL-4B at 128/256/512 frames while meaningfully reducing frames or latency.
  2. Report computational cost and latency. Although VideoBrain uses fewer frames, it requires sequential MLLM inference and tool calls, which may increase end-to-end latency.

- **Data quality is limited by the teacher model.** The quality of both the reasoning trajectories and the question-category classification appears heavily constrained by the teacher model’s original tool-use capability.

- **Rigid agent hyperparameters.** At inference time, the CLIP Agent and Uniform Agent are fixed to return 4 and 8 frames per call, respectively, with a maximum of 5 interaction rounds. Such static allocation may be suboptimal for videos with very high or very low information density.

- **Limited CLIP retrieval search space.** The CLIP Agent retrieves from at most 256 uniformly sampled candidate frames. For hour-long videos with tens of thousands of frames, this search space is still sparse and may miss brief but critical moments.

---

> ### Author Rebuttal · Authors · 2026-03-31
>
> We thank the reviewer for the thorough evaluation. We address each concern below.
>
> **[W1] Higher-Frame Experimental Setting.**
> We provide a higher-frame comparison using Qwen3-VL-8B-Instruct as the uniform sampling baseline (256f) and VideoBrain initialized with 128 frames (init=128f):
>
> | Model | LVBench | Video-MME Long | LongVideoBench | Avg Frames |
> |-------|---------|----------------|----------------|------------|
> | Qwen3-VL-8B Baseline (256f) | 40.4 | 51.4 | 50.2 | 256 |
> | **VideoBrain (init=128f)** | **46.5** | **52.5** | **57.6** | **131.5** |
>
> VideoBrain uses on average **131.5 frames**—roughly half of the 256-frame baseline—while achieving substantial accuracy gains (+6.1% / +1.1% / +7.4%), showing that adaptive sampling remains beneficial even at higher frame budgets. Moreover, scaling to very high frame counts quickly becomes impractical due to memory costs. Non-weight runtime memory grows approximately linearly with frame count: at 256 frames it is already several GB, and at 2048 frames it becomes substantially larger, making such settings significantly more resource-intensive and less practical under limited GPU memory.
>
> **[W2] Computational Cost and Latency.**
> We use SigLIP2-ViT-B/16 as the CLIP encoder, which is lightweight. Processing 256 frames requires $\approx$5 TFLOPs, introducing small computational overhead.
>
> | Method | FLOPs | Avg Running Time† | Weights |
> |--------|-------|-----------------|---------|
> | SigLIP2 (256f) | $\approx$5 TFLOPs | 3.1s | $\approx$0.3 GB (FP32) |
>
> † Measured concurrently with VideoBrain inference on the same GPU, reflecting the actual deployment scenario.
>
> The following table presents End-to-end latency on LVBench, measured on a single H20 GPU. To ensure a fair comparison, the CLIP encoder and VideoBrain share the same GPU during inference, accounting for the additional memory overhead introduced by CLIP. VideoBrain outperforms 256-frame inference in both accuracy (**41.3%** vs. 40.4%) and speed (**30.5s** vs. 90.4s). The higher latency vs. the 32-frame baseline is due to reasoning mode and multi-turn interaction.
>
> | Method | Acc (%) | Frames | Latency |
> |--------|---------|--------|---------|
> | Qwen3-VL-8B | 32.3 | 32 | 5.1s |
> | Qwen3-VL-8B | 40.4 | 256 | 90.4s |
> | **VideoBrain** | **41.3** | **21.6** | **30.5s** |
>
> **[W3] Data Quality Limited by Teacher Model.**
> We acknowledge that SFT trajectories are generated by the teacher model. However, the RL stage further optimizes the policy through direct environment interaction, independent of teacher quality. Removing RL causes clear accuracy drops, demonstrating that RL refines the model substantially beyond what SFT alone achieves:
>
> | | LVBench | Video-MME Long |
> |--|---------|----------------|
> | w/ RL (VideoBrain) | 41.3 | 49.8 |
> | w/o RL | 39.1 | 46.9 |
> | $\Delta$ | $-$2.2 | $-$2.9 |
>
> **[W4] Rigid Agent Hyperparameters.**
> We analyzed invocation patterns across 2,298 samples:
>
> | Rounds | Accuracy | Sample Ratio |
> |--------|----------|--------------|
> | 1 | 49.9% | 62.6% |
> | 2 | 44.0% | 32.9% |
> | 3 | 47.7% | 2.8% |
> | 4 | 40.0% | 0.4% |
> | 5 | 25.0% | 0.3% |
>
> We find that **95.5\%** of questions can beare resolved within 2 rounds. Accuracy decreases at higher rounds because these are inherently harder questions (the slight rise at round 3 reflects small-sample fluctuation at only 2.8%). Capping at 5 rounds prevents wasteful computation on the remaining 3.5% that cannot be resolved regardless of additional frames. The fixed 4/8 frames per call reflects a deliberate efficiency trade-off, as each call targets a specific temporal interval rather than a global search.
>
> **[W5] Limited CLIP Search Space.**
> Our results suggest that increasing the number of candidate frames could further improve performance. Determining the optimal search space is an interesting direction for future work. For this work, we focus on demonstrating the functionality and effectiveness of the proposed framework, and leave a systematic exploration of candidate frame counts to future work.

---

> > ### Author Rebuttal · Reviewer_EXmM · 2026-04-04
> >
> > Thank you for the response. My concern has been fully addressed. I suggest that the authors incorporate these discussions into the final version.

---

### Official Review · Reviewer_ZKsh · 2026-03-12

**Soundness:** 3
**Presentation:** 3
**Significance:** 3
**Originality:** 3
**Overall Recommendation:** 4
**Confidence:** 3

**Summary:**

### Strengths

1. The dual-agent design is well-motivated and addresses a real limitation of prior work. Combining semantic retrieval (CLIP agent) with temporal sampling (Uniform agent) provides complementary capabilities that cover different question types. The ablation study confirms both agents contribute meaningfully.

2. The behavior-aware reward function is a principled solution to the reward hacking problem. The Direct/Adaptive/Active classification with category-specific rewards explicitly prevents models from invoking agents indiscriminately. This design is clearly explained.

3. The paper evaluates on four long video benchmarks with consistent improvements, provides thorough ablation studies covering agents, training stages, dataset classification, and reward components. The generalization experiment to short video benchmarks further strengthens the findings.

### Weaknesses

1. All experiments use Qwen3-VL-8B as the base model and Qwen3-VL-235B as the teacher. The generalizability to other VLM families (e.g., InternVL, LLaVA-OneVision) remains unclear.

2. This work relies on a 235B teacher model for data classification. The computational cost of this preprocessing step is not quantified, making it difficult to assess the true overhead of the pipeline.

3. The paper lacks failure analysis (e.g., when and why the method fails). It would be valuable to understand the error patterns, especially considering that only 20-22 frames are typically utilized for hour-long videos on average.

4. FrameThinker uses Qwen2.5-VL-7B while VideoBrain uses Qwen3-VL-8B. The performance gap may partly reflect backbone improvements rather than methodological advantages.

**Compliance With Llm Reviewing Policy:**

Affirmed.

**Final Justification:**

I appreciate the authors' response and maintain my positive rating of Weak Accept.

**Key Questions For Authors:**

Q1. Backbone generalization
Have you evaluated VideoBrain with other VLM backbones? The current evaluation is limited to the Qwen3-VL family for both the base and teacher models.

Q2. Teacher model dependency
How sensitive is the data classification quality to the choice of teacher model? Would using a smaller teacher (e.g., Qwen3-VL-32B) significantly degrade the quality of Direct/Adaptive/Active labels?

Q3. Failure analysis
Could you provide examples of failure cases, particularly where the model invokes agents but still fails? Understanding the failure modes would help assess the method's limitations.

Q4. Fair comparison with FrameThinker
Since FrameThinker uses Qwen2.5-VL-7B and VideoBrain uses Qwen3-VL-8B, could you provide a comparison using the same backbone to isolate the contribution of the proposed method from the backbone improvement?

**Limitations:**

The paper acknowledges the dependency on the teacher model and the computational cost of data classification. However, it does not discuss failure modes, the scalability of the approach to even longer videos (e.g., multi-hour), or potential biases introduced by the CLIP model in semantic retrieval.

**Strengths And Weaknesses:**

### Strengths

1. The dual-agent design is well-motivated and addresses a real limitation of prior work. Combining semantic retrieval (CLIP agent) with temporal sampling (Uniform agent) provides complementary capabilities that cover different question types. The ablation study confirms that both agents contribute meaningfully.

2. The behavior-aware reward function is a principled solution to the reward hacking problem. The Direct/Adaptive/Active classification with category-specific rewards explicitly prevents models from invoking agents indiscriminately. This design is well thought out and clearly explained.

3. Comprehensive experimental evaluation. The paper evaluates on four long video benchmarks with consistent improvements, provides thorough ablation studies covering agents, training stages, dataset classification, and reward components. The generalization experiment to short video benchmarks further strengthens the findings.

4. The paper achieves improvements while using fewer frames (30-40% reduction), which is a practical advantage for real-world deployment where computational efficiency matters.

### Weaknesses

1. Single backbone evaluation. All experiments use Qwen3-VL-8B as the base model and Qwen3-VL-235B as the teacher. The generalizability to other VLM families (e.g., InternVL, LLaVA-OneVision) remains unclear.

2. Heavy reliance on a 235B teacher model for data classification. The computational cost of this preprocessing step is not quantified, making it difficult to assess the true overhead of the pipeline.

3. Lack of failure analysis. The paper does not discuss when and why the method fails. Given that only 20-22 frames are used on average for hour-long videos, understanding error patterns would be valuable.

4. Unfair comparison with FrameThinker. FrameThinker uses Qwen2.5-VL-7B while VideoBrain uses Qwen3-VL-8B. The performance gap may partly reflect backbone improvements rather than methodological advantages.

---

> ### Author Rebuttal · Authors · 2026-03-31
>
> We thank the reviewer for the positive assessment and constructive feedback. We address each concern below.
>
> **[W1/Q1] Backbone Generalization.**
> We evaluate VideoBrain on **Qwen2.5-VL-7B** as an additional backbone. Results show consistent improvements across benchmarks, demonstrating that the framework generalizes beyond the Qwen3-VL family:
>
> | Model | Params | LVBench | Video-MME Long | LongVideoBench |
> |-------|--------|---------|----------------|----------------|
> | Qwen2.5-VL-Instruct | 7B | 31.6 | 41.9 | 43.2 |
> | **VideoBrain (Qwen2.5-VL)** | 7B | **38.7** | **48.3** | **50.3** |
> | Qwen3-VL-Instruct | 8B | 32.3 | 45.6 | 45.2 |
> | **VideoBrain (Qwen3-VL)** | 8B | **41.3** | **49.8** | **53.3** |
>
>
> **[W2/Q2] Teacher Model Dependency.**
> Qwen3-VL-235B-A22B is a MoE model that activates only 22B parameters during inference, making it significantly more efficient than a dense 235B model. We investigate the sensitivity of data classification to the choice of teacher model by comparing Qwen3-VL-235B-A22B (ours) with a smaller Qwen3-VL-32B teacher. A key metric is the proportion of samples where the base model fails but the teacher succeeds **via agent invocation** (i.e., the Active category with high-quality trajectories):
>
> | Teacher Model | Direct | Adaptive | Active | Agent-Correct Ratio |
> |---------------|--------|----------|--------|---------------------|
> | Qwen3-VL-32B | 38.2% | 13.7% | 48.1% | 10.9% |
> | Qwen3-VL-235B-A22B | 41.5% | 11.7% | 46.8% | **17.2%** |
>
> Our SFT training uses both Adaptive samples (base wrong, teacher correct without agent) and Active samples with agent-success trajectories (the Agent-Correct Ratio column). The 32B teacher produces significantly fewer agent-success trajectories (10.9% vs 17.2%), creating a data imbalance in the SFT set: fewer examples demonstrating effective agent invocation. This directly impacts model performance: as shown below, training with the 32B teacher leads to lower accuracy **and** reduced agent usage (fewer frames), confirming that the model learns less effective agent behavior:
>
> | Teacher Model | LVBench | Video-MME Long | Avg Frames |
> |---------------|---------|----------------|------------|
> | Qwen3-VL-32B | 36.2 | 44.8 | 19.2 |
> | **Qwen3-VL-235B-A22B** | **39.1** | **46.9** | **20.7** |
>
> Both experiments use SFT training only (without RL) to isolate the effect of teacher quality. These results confirm that teacher quality is important for generating high-quality Active trajectories, and that Qwen3-VL-235B-A22B is a more appropriate choice for our pipeline.
>
> **[W3/Q3] Failure Analysis.**
> We analyze failure cases where VideoBrain invokes agents but still fails to answer correctly. We manually annotated 70 of such cases and categorized each by error type.
>
> | Error Type | Ratio |
> |------------|-------|
> | Imprecise Agent Sampling | 54.3% |
> | Visual Hallucination | 20.0% |
> | Incomplete Exploration | 17.1% |
> | Failed Answer Grounding | 5.7% |
> | Wrong Tool Selection | 2.9% |
>
> **Imprecise Agent Sampling** has two sub-causes: (1) incorrect frame range specification: the model selects an interval that is too wide or inadvertently excludes the answer frame; (2) imprecise CLIP prompt: the text query does not accurately capture the visual semantics. For example, for the question *"What is the first warning on the warning sign in the volcano skali?"*, the model issued a CLIP query of *"volcano Skali warning sign lava"*, whereas a more precise query such as *"warning notice"* would have retrieved the relevant frame more effectively. Overly verbose prompts dilute the semantic signal and reduce the retrieval precision.
>
> **Incomplete Exploration** (17.1%) reflects cases where the model terminates reasoning too early. While 62.6% of questions are answered in round 1, some harder questions require additional agent calls to achieve meaningful accuracy improvement, suggesting that the model's difficulty awareness is not yet perfect. We regard both error types as directions for future work, including better agent prompt generation and more adaptive round-allocation strategies.
>
> **[W4/Q4] Fair Comparison with FrameThinker.**
> We provide a same-backbone comparison using Qwen2.5-VL-7B for both methods:
>
> | Model | LVBench | Frames | Video-MME Long | Frames | LongVideoBench | Frames |
> |-------|---------|--------|----------------|--------|----------------|--------|
> | FrameThinker (Qwen2.5-VL) | 36.6 | 23.9 | 47.6 | 23.9 | 52.9 | 21.1 |
> | **VideoBrain (Qwen2.5-VL)** | **38.7** | **19.6** | **48.3** | **18.3** | 50.3 | **19.3** |
>
> VideoBrain outperforms FrameThinker on LVBench (+2.1%) and Video-MME Long (+0.7%) with fewer frames. On LongVideoBench, FrameThinker scores slightly higher (52.9 vs 50.3), while VideoBrain uses fewer frames (19.3f vs 21.1f), achieving a comparable result more efficiently. Overall, these results confirm that the performance gains in our main paper reflect genuine methodological improvements beyond backbone differences.

---

> > ### Author Rebuttal · Reviewer_ZKsh · 2026-04-02
> >
> > The authors have adequately addressed my major concerns in the rebuttal.
> >
> > In particular, the additional experiments on backbone generalization and the same-backbone comparison provide convincing evidence of the method’s robustness. I especially appreciate the detailed failure case analysis, which offers valuable insights into the model’s behavior and limitations, and strengthens my confidence in the work.
> > Overall, I am satisfied with the responses and maintain my positive assessment.

---

### Official Review · Reviewer_mxAT · 2026-03-12

**Soundness:** 3
**Presentation:** 3
**Significance:** 2
**Originality:** 2
**Overall Recommendation:** 4
**Confidence:** 3

**Summary:**

This paper presents a novel framework, VideoBrain, which applies agentic reinforcement learning to Video MLLMs and designs a reward function to avoid bonus hacking. By using a CLIP agent and a Uniform Agent, VideoBrain achieves improvements of +3.5% to +9.0% over long-video benchmarks while using 30–40% fewer frames.

**Compliance With Llm Reviewing Policy:**

Affirmed.

**Final Justification:**

I will maintain my positive score

**Key Questions For Authors:**

1. I am curious how the method would perform on datasets such as VideoMMMU, which involve less general scenarios where CLIP may not handle the semantics effectively.

**Limitations:**

1. Calling the CLIP Agent may introduce additional computational overhead.
2. Performance on out-of-domain videos may also be limited due to the reliance on CLIP.

**Strengths And Weaknesses:**

## Strength
1. The visualization results appear promising; the model trained with RL is able to correctly utilize the agent tools.
2. The semantic and temporal retrieval tool adopted in VideoBrain is intuitive and well motivated.
3. The method shows promising results on the benchmarks.
4. The paper is clearly written and easy to follow.
## Weakness
1. VideoBrain is highlighted for its lower frame count, which is intended to reduce computational cost. However, the CLIP agent needs to process 128–256 frames per video, which may significantly increase the overall computation. I believe an overall analysis of computational efficiency is needed.
2. The ablation study may include results on more datasets rather than only LVBench.
3. Despite the promising results on Qwen3-VL-8B-Instruct, implementing the framework on other base models (e.g., Qwen2.5-VL, which has weaker native agentic abilities) may also help facilitate further research.

---

> ### Author Rebuttal · Authors · 2026-03-31
>
> We thank the reviewer for the constructive feedback. We address each concern below.
>
> **[W1] Computational Efficiency.**
> We use SigLIP2-ViT-B/16 as the CLIP encoder, which is lightweight. Processing 256 frames requires $\approx$5 TFLOPs, introducing small computational overhead.
>
> | Method | FLOPs | Avg Running Time† | Weights |
> |--------|-------|-----------------|---------|
> | SigLIP2 (256f) | $\approx$5 TFLOPs | 3.1s | $\approx$0.3 GB (FP32) |
>
> † Measured concurrently with VideoBrain inference on the same GPU, reflecting the actual deployment scenario.
>
> The following table presents End-to-end latency on LVBench, measured on a single H20 GPU. To ensure a fair comparison, the CLIP encoder and VideoBrain share the same GPU during inference, accounting for the additional memory overhead introduced by CLIP. VideoBrain outperforms 256-frame inference in both accuracy (**41.3%** vs. 40.4%) and speed (**30.5s** vs. 90.4s). The higher latency vs. the 32-frame baseline is due to reasoning mode and multi-turn agent interaction.
>
> | Method | Acc (%) | Frames | Latency |
> |--------|---------|--------|---------|
> | Qwen3-VL-8B | 32.3 | 32 | 5.1s |
> | Qwen3-VL-8B | 40.4 | 256 | 90.4s |
> | **VideoBrain** | **41.3** | **21.6** | **30.5s** |
>
> **[W2] Ablation on More Datasets.**
> We extend ablations to Video-MME Long and MLVU Test. The conclusions from LVBench hold consistently across all three benchmarks.
>
> | Model | LVBench | Video-MME Long | MLVU Test |
> |-------|---------|----------------|-----------|
> | Baseline | 32.3 | 45.6 | 43.4 |
> | **VideoBrain** | **41.3** | **49.8** | **46.9** |
> | w/o CLIP Agent | 36.3 | 47.0 | 40.3 |
> | w/o Uniform Agent | 39.8 | 46.9 | 44.9 |
> | w/o RL | 39.1 | 46.9 | 46.6 |
> | w/o SFT | 35.5 | 45.1 | 37.5 |
> | w/o SFT Classified Data | 36.2 | 45.4 | 46.3 |
> | w/o RL Classified Data | 40.8 | 49.1 | 46.4 |
> | w/o Reward Shaping | 39.0 | 46.6 | 46.4 |
> | w/o Format Check | 39.7 | 48.2 | 44.1 |
>
> **[W3] Other Base Models.**
> We evaluate on **Qwen2.5-VL-7B** as an additional backbone. VideoBrain consistently improves Qwen2.5-VL-7B by +7.1% / +6.4% / +7.1% across three benchmarks, demonstrating that the framework is effective regardless of the backbone capability.
>
> | Model | Params | LVBench | Video-MME Long | LongVideoBench |
> |-------|--------|---------|----------------|----------------|
> | Qwen2.5-VL-Instruct | 7B | 31.6 | 41.9 | 43.2 |
> | **VideoBrain (Qwen2.5-VL)** | 7B | **38.7** | **48.3** | **50.3** |
> | Qwen3-VL-Instruct | 8B | 32.3 | 45.6 | 45.2 |
> | **VideoBrain (Qwen3-VL)** | 8B | **41.3** | **49.8** | **53.3** |
>
> **[Q1] Performance on VideoMMMU.**
> We evaluate on VideoMMMU, which covers specialized domain knowledge (Art, Business, Science, Medicine, Humanities and Engineering). VideoBrain achieves **+2.1%** improvement with **47% fewer frames**. This shows that our dual-agent design remains effective even in domain-specialized scenarios. Furthermore, VideoMMMU has an average video duration of **506.2 seconds** (~8 minutes), placing it in the medium-to-short video range. In this setting, VideoBrain learns to answer many questions directly from fewer frames without unnecessary agent invocation, which explains the significant frame reduction (-47%) while still improving accuracy.
>
> | Model | Acc (%) | Frames |
> |-------|---------|--------|
> | Qwen3-VL-8B | 65.3 | 32 |
> | **VideoBrain** | **67.4** | **17.1** |
> | $\Delta$ | **+2.1** | **-47%** |

---

> > ### Author Rebuttal · Reviewer_mxAT · 2026-04-03
> >
> > Thanks for the detailed rebuttal. The newly provided evidence and explanations well addressed my concerns. Therefore, I will maintain my positive score.

---

### Decision · Program_Chairs · 2026-04-30

**Decision:**

Accept (regular)

**Comment:**

This paper proposes an end-to-end framework for long-video understanding that adaptively samples frames with two complementary agents trained with a behavior-aware reward function. The dual-agent design is well-motivated and the behavior-aware reward offers question-aware solutions. Also, the empirical results are consistent and compelling with about 3% ~ 9% accuracy gains using 30%~40% fewer frames. The rebuttal effectively addressed the issues over generalization on Qwen2.5-VL-7B as the baseline, FrameThinker and failure case analysis. All reviewers maintained or raised their positive ratings after the rebuttal, highlighting the clarity of the paper, the strong motivation, and the practical efficiency gains. Considering the strong reviewers’ support and the significance of the proposed framework, I recommend this paper for acceptance.